# 3D Crustal Structure of the Ligurian Basin Revealed by Surface Wave Tomography using Ocean Bottom Seismometer Data

Felix Noah Wolf[1], Dietrich Lange[1], Anke Dannowski[1], Martin Thorwart[2], Wayne Crawford[3], Lars Wiesenberg[2], Ingo Grevemeyer[1], Heidrun Kopp[1,2], and the AlpArray Working Group*

[1] GEOMAR Helmholtz Centre for Ocean Research Kiel, Kiel, 24148, Germany
[2] Institute of Geosciences, Kiel University, Kiel, 24118, Germany
[3] Laboratoire de Géosciences Marines, Institut de Physique du Globe de Paris, Paris, 75238 Cedex 5, France
*www.alparray.ethz.ch/

*Correspondence to*: Felix N. Wolf (fnwolf@geomar.de)

**Abstract.** The Liguro-Provençal basin was formed as a back-arc basin of the retreating Calabrian-Apennines subduction zone during the Oligocene and Miocene. The resulting rotation of the Corsica-Sardinia block is associated with rifting, shaping the Ligurian Basin. It is still debated whether oceanic or atypical oceanic crust was formed or if the crust is continental and experienced extreme thinning during the opening of the basin. We perform ambient noise tomography, also taking into account teleseismic events, using an amphibious network of seismic stations, including 22 broadband Ocean Bottom Seismometers (OBS), to investigate the lithospheric structure of the Ligurian Basin. The instruments were installed in the Ligurian Basin for eight months between June 2017 and February 2018 as part of the AlpArray seismic network. Because of additional noise sources in the ocean, OBS data are rarely used for ambient noise studies. However, we carefully pre-process the data, including corrections for instrument tilt and seafloor compliance and excluding higher modes of the ambient-noise Rayleigh waves. We calculate daily cross-correlation functions for the AlpArray OBS array and surrounding land stations. We also correlate short time windows that include teleseismic earthquakes, allowing us to derive surface wave group velocities for longer periods than using ambient noise only. We obtain group velocity maps by inverting Green's functions derived from the cross-correlation of ambient noise and teleseismic events, respectively. We then used the resulting 3D group velocity information to calculate 1D depth inversions for S-wave velocities. The group velocity and shear-wave velocity results compare well to existing large-scale studies that partly include the study area. Onshore France, we observe a high-velocity area beneath the Argentera Massif, roughly 10 km below sea level. We interpret this as the root of the Argentera Massif. Our results add spatial resolution to known seismic velocities in the Ligurian Basin, thereby augmenting existing seismic profiles. In agreement with existing seismic studies, our shear-wave velocity maps indicate a deepening of the Moho from 12 km at the southwestern basin centre to 20-25 km at the Ligurian coast in the northeast and over 30 km at the Provençal coast. The maps also indicate that the southwestern and northeastern Ligurian Basin are structurally separate. The lack of high crustal $v_P/v_S$ ratios beneath the southwestern part of the Ligurian Basin preclude mantle serpentinisation there.

# 1 Introduction

The Ligurian Basin is a marginal basin located in the north-western Mediterranean Sea at the transition from the Alpine orogen to the Apennine system (Fig. 1). It formed as a back-arc basin to the southeastward trench retreat of the Apennines-Calabrian subduction zone during the late Oligocene and Miocene (Gueguen et al., 1998; Rollet et al., 2002). Rifting in the Liguro-Provençal basin initiated about 32 Ma ago (Jolivet et al., 2015). From 21 Ma, the rifting was followed by a counter-clockwise rotation of the Corsica-Sardinia block by approximately 30 degrees (e.g. Vigliotti and Langenheim, 1995; Jolivet and

Faccenna, 2000; Rollet et al., 2002; Speranza et al., 2002; Schettino and Turco, 2006). Gattacceca et al. (2007) estimate a rotation of 45 degrees, based on $^{40}Ar/^{39}Ar$ geochronological investigations of Miocene volcanic sequences in Sardinia. Le Breton et al. (2017) describe a total amount of counter-clockwise rotation of the Corsica-Sardinia block by at least 53 degrees during the last 35 Ma. At the end of the Burdigalian Age (about 16-18 Ma), the Corsica-Sardinia block was stranded between the Apennines and the European margin in southern France (Rosenbaum et al., 2002). The opening of the Ligurian Basin

terminated, while the roll-back of the Calabrian subduction zone continued and initiated the opening of the Tyrrhenian Sea (e.g. Faccenna et al., 2001). Today, the Ligurian Basin is 150-225 km wide (Dannowski et al., 2020), broadening from the northeast to the southwest. The continental margin is narrow (10-20 km) and steep along the Ligurian coast (Finetti et al., 2005) and broader (20-50 km) on the Corsican side (e.g. Rollet et al., 2002). The marine bedrock is covered by a sedimentary layer (e.g. Schettino and Turco, 2006) of variable thickness: less than 3 km thick near the Tuscany coast, increasing towards

the southwest to a thickness of up to 8 km offshore Marseille. Rollet et al. (2002) identify several areas of magmatic intrusions related to three phases of calcalkaline and alkaline volcanism. The first is linked to the opening of the basin, the second links to the transition of the Calabrian subduction zone to the Tyrrhenian Sea, and the third mainly occurred north of Corsica and in the Gulf of Genova (12-11 Ma).

The crust-mantle-boundary is well defined along several seismic lines (detailed overview in Dannowski et al., 2020), but

otherwise only estimated in parts from large-scale surface wave studies (e.g. Molinari et al., 2015b; Kästle et al., 2018; Lu et al., 2018). Parallel to the basin, Dannowski et al. (2020) explain the satellite-derived free-air anomaly (Sandwell et al., 2014) by gravity modelling along their refraction seismic line (Fig. 1). Dannowski et al. (2020) also include the directly connecting wide-angle reflection seismic line by Makris et al. (1999, Fig. 1). Both seismic and gravity modelling reveal similar values for the Moho depth, showing a gradual thickening of continental crust towards the northeast. At the southwestern end of the

seismic refraction profile, the Moho is about 12 km deep. It gradually deepens towards the northeast, reaching a depth of 22 km close to the Italian coast. Contrucci et al. (2001) estimated the Moho depth along the multichannel seismic line LISA01 (Fig.1). They observe a decrease in Moho depth from 18 km at the Corsican margin to 13 km in the basin centre and an increase to over 30 km towards the Provençal coast. This variability is supported by the surface wave derived Moho map of Kästle et al. (2018), showing an increasing Moho depth from the Ligurian Basin (< 20 km) towards the coast (> 25 km).

Many studies addressed whether continental crust was extremely thinned during the rifting or if oceanic spreading occurred and formed oceanic crust in the basin centre. Several authors (Rollet et al., 2002; Gailler et al., 2009; Jolivet et al., 2015)

propose an area of atypical oceanic crust, characterised as being very thin (< 4 km) and showing complex magnetic anomalies that cannot be correlated to isochrons (e.g. Rollet et al., 2002; Schettino and Turco, 2006), in the basin-centre. This area is located next to a broad transition zone towards continental crust at the basin's edges. Based on a recent refraction seismic study, Dannowski et al. (2020) propose that seafloor spreading did not occur during the formation of the Ligurian Basin. They show that beneath the southwestern part of the basin, the continental crust thins and possibly gives way to partly serpentinised mantle lying directly beneath an up to 7 km thick sediment cover. Schettino & Turco (2006) find a similar sediment thickness based on a joint interpretation of magnetic and seismic data.

Another open question relates to the location of the prolongation of the Alpine front. It is well defined onshore France and Corsica, but it remains unclear if and where the connection of these parts of the Alpine front is preserved offshore. At the scale of the entire Alpine belt region, land data based ambient noise tomographies have been performed by (Molinari et al., 2015b; Kästle et al., 2018; and Lu et al., 2018). These studies revealed the onshore geometry but did not cover the Ligurian Basin. Guerin et al. (2020) conducted an ambient noise surface-wave tomography study along the southwestern Alps and a small part of the Ligurian margin using five ocean bottom seismometers (OBS) and two offshore cabled seismometers close to Nice. They identify a low-velocity zone offshore Nice that is linked to salt and evaporite deposits.

To better understand the present-day crustal velocity structure and its implications on the evolution of the Ligurian Basin, we use a unique amphibious seismic network covering the entire Ligurian Basin and adjacent coastal areas, providing high-resolution group velocity maps and a three-dimensional shear velocity model.

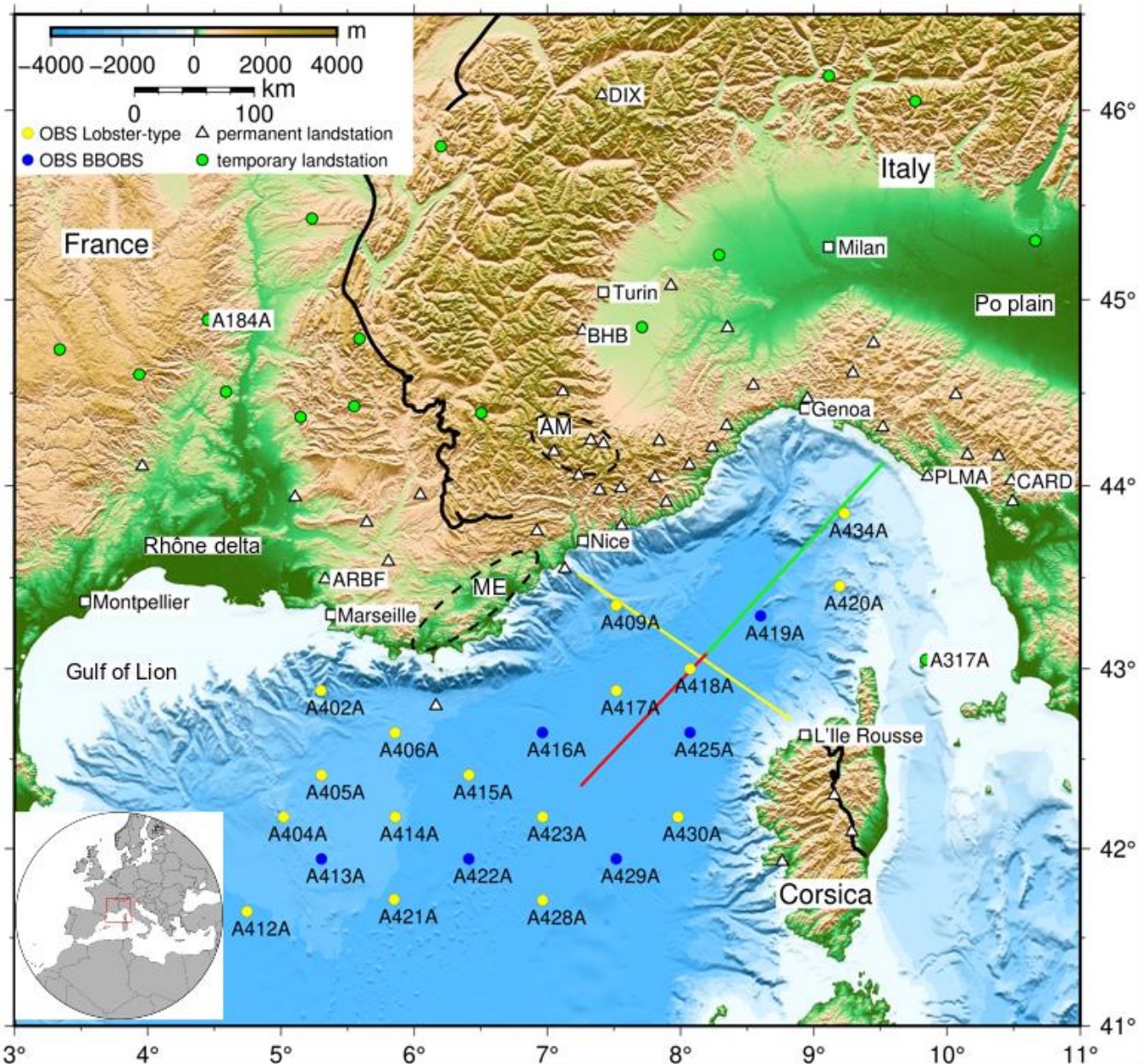

**Figure 1: Map of the Ligurian Basin and adjacent Alpine region and the stations used for this study. OBS stations (network code: Z3) are shown as yellow (Lobster-type) and blue (BBOBS) circles. Permanent land stations (network codes: CH, FR, GU, IV, and MN, see Table S1) are shown as white triangles. Temporary land stations from AlpArray (network code: Z3) are shown as green circles. Station names are given for the OBS and land stations mentioned in the text or used in Fig. 3. White squares represent cities. The black line represents the Alpine front (Schmid et al., 2004), AM marks the Argentera Massif, and ME marks the Maures-Esterel Massif. The inlay map in the bottom left shows the location of the research area (red box). The red, green, and yellow lines show**

seismic refraction and reflection lines; red: Dannowski et al. (2020), green: Makris et al. (1999), yellow: LISA01, Contrucci et al. (2001). The topography is plotted based on a GMRT grid (Ryan et al., 2009).

## 2 Data

A network of 22 broadband ocean-bottom seismometers (OBS) was installed jointly by the *Institut de physique du globe de Paris* (IPGP, Paris, France), the *Institut des Sciences de la Terre* (ISTerre, Grenoble, France) and *GEOMAR Helmholtz Centre for Ocean Research Kiel* (Kiel, Germany) (Fig. 1) to investigate the velocity structure of the crust and upper mantle beneath the Ligurian Basin. The AlpArray OBS array is the offshore component of the AlpArray seismic network (Hetényi et al., 2018). The instruments were deployed from the RV *Pourquoi Pas?* in June 2017 and were recovered in February 2018 by RV

*Maria S. Merian*.

The AlpArray OBS network consisted of six French OBS (BBOBS) and 16 OBS (Lobster-Type) provided by the German instrument pool (DEPAS, Schmidt-Ausch and Haberland, 2017). The BBOBS were equipped with three-component Nanometrics Trillium 240 broadband seismometers with a lower corner period of 240 s and a differential pressure gauge (DPG) designed by the Scripps Institution of Oceanography (Cox et al., 1984). The sampling rate of the installed LCHEAPO recorder

was 62.5 Hz. The DEPAS OBS were equipped with Trillium compact seismometers by Nanometrics Inc. with a lower corner period of 120 s and HTI-01-PCA hydrophones from High Tech Inc. The sampling rate of the K.U.M. 6D6 recorder was 250 Hz.

The instrument clocks were synchronised with GPS time before deployment and after recovery to reveal any internal clock drift and apply a linear clock drift correction. We calculated every station's probabilistic power spectral densities (PPSDs)

(McNamara and Buland, 2004). The lowest spectral levels on the vertical seismometer components fall in between the mean minimum and maximum noise levels for land stations (Peterson, 1993) for both the German (Fig. S1 a-b) and French OBS (Fig. S1 c-d). Regarding the pressure component, the French DPGs yield high-quality data (-38 dB to 40 dB) while the HTI hydrophones have a range of -20 dB to 40 dB with a lesser resolution for periods larger than 10 s (Fig. S1 a, c). These results are comparable to similar instrument setups (Stähler et al., 2016) used during the RHUM-RUM OBS experiment in the Indian

Ocean. To resolve the entire structure of the Ligurian Basin and the surrounding areas onshore, we incorporated 16 temporary and 42 permanent land stations in our analysis (Table S1).

## 3 Methods

The ambient noise technique was developed during the last 20 years (e.g. Lobkis & Weaver, 2001; Wapenaar et al., 2010a; Wapenaar et al., 2010b) and is based on the concept of Aki (1957) regarding the spectra of stationary stochastic waves. Ambient

noise techniques exploit the 'noise' of long-term recordings as the desired signal. This part of the measured signal includes,

for example, anthropogenic noise, microseismic signals from ocean-coast interactions, and highly scattered waves of teleseismic origin (Campillo and Paul, 2003; Campillo and Roux, 2016). Given a continuous measurement and uniformly distributed noise sources, the cross-correlation of recordings of two stations is used as the empirical Green's function representing the subsurface response to a wave propagating from one station to the other. These empirical Green's functions from different station pairs are used to invert for two-dimensional (2D) group velocity maps, 1D velocity-depth profiles or 3D velocity distribution maps.

Although the technique is well established for land data (e.g. Barmin et al., 2001; Campillo & Paul, 2003; Shapiro et al., 2005; Prieto et al., 2009; Goutorbe et al., 2015; Kästle et al., 2019), it is not yet used regularly for ambient noise analysis on ocean bottom seismometer data. Previous studies show that ambient noise can be calculated using OBS data (e.g. Harmon et al., 2007, 2012; Takeo et al., 2014; Dewangan et al., 2018). However, compared to land stations, seismic recordings on OBS contain less anthropogenic noise but other additional noise sources like tilt and compliance noise (Crawford et al., 1998; Webb, 1998; Bell et al., 2015).

### 3.1 Pre-processing - tilt and compliance correction

Adimah & Padhy (2020) showed that reducing tilt and compliance noise before running the cross-correlation proves beneficial, as tilt and compliance noise are not part of the useful signal. Therefore, we pre-process the OBS data as proposed by Crawford & Webb (2000). First, we cut the continuous OBS recordings into daily files and resample the data at 1 Hz. Next, we remove tilt and compliance noise. Tilt noise is introduced by a slight inclination of the instrument, causing horizontal movement to appear on the vertical component (Crawford & Webb, 2000). Although the instruments level themselves to an accuracy of ±0.5° (Lobster-Type) and ±5° (BBOBS), respectively, the remaining tilt is sufficient to create tilt noise. The tilt of the instrument can be caused by processes such as ocean bottom currents. On the other hand, compliance is a signal generated by ocean infragravity waves introducing pressure fluctuations that cause µm-scale deformation of the seafloor (Webb and Crawford, 1999). The variations of the gravitational forces of the water column, the deformation of the seafloor itself, and the caused variation in the distance of the OBS to the Earth's gravitational centre all introduce changes to the measured acceleration (Crawford et al., 1998). Thus, compliance increases vertical acceleration noise level by 10 dB to 25 dB for 30-100 s (Webb and Crawford, 1999).

To correct for tilt and compliance noise, we applied the procedure described in Crawford & Webb (2000) and Bell et al. (2015). First, we calculate a transfer function between the vertical seismometer component and the hydrophone component. Next, we subtract the coherent part of the signal (in this case: compliance) from the vertical seismometer component. We also corrected both horizontal components for compliance before removing tilt noise (Crawford & Webb, 2000). Subsequently, the same routine is used to remove the coherent signal between the vertical component and each horizontal component to remove tilt noise. Thus, we calculate the transfer functions between the vertical component and each of the horizontal components. Finally, we obtain a vertical component corrected for tilt and compliance. The order in which the components are corrected is interchangeable. The land station recordings were not corrected for tilt and compliance noise but are also resampled to 1 Hz.

## 3.2 Ambient noise technique – cross-correlation and mode identification

*Cross-correlation*

We use the tilt- and compliance-corrected daily files to estimate cross-correlation functions (CCF) for every vertical component OBS-OBS and OBS-land station pair (Bensen et al., 2007). Additionally, we calculate CCFS for all land-land pairs for the land stations A317A, ARBF, and DIX (see Fig. 1) and CCFs for all combinations of 20 selected land stations (namely AJAC, BLAF, BOB, BSTF, CALF, CARD, EILF, ENR, GBOS, IMI, ISO, MSSA, PCP, PLMA, ROTM, SAOF SMPL, TRBF, TURF,

and VLC) to increase the ray coverage onshore (Fig. 2). The cross-correlation is calculated day-wise for every station pair. Afterwards, we stack the single-day CCFs to estimate one CCF per station pair.

In addition to ambient noise cross-correlations, we correlate time windows (45 min long) that include strong teleseismic events using the two-station method (e.g. Meier et al., 2004; Boschi et al., 2013; Tonegawa et al., 2020). We only use station pairs for which the stations' azimuth equals the great circle from the event to within ± 7 °. The correlation window, starting at the

origin time of the event, is dominated by the earthquake signals. The further processing is identical to correlating ambient noise day files but is performed for longer periods (20 s to 90 s).

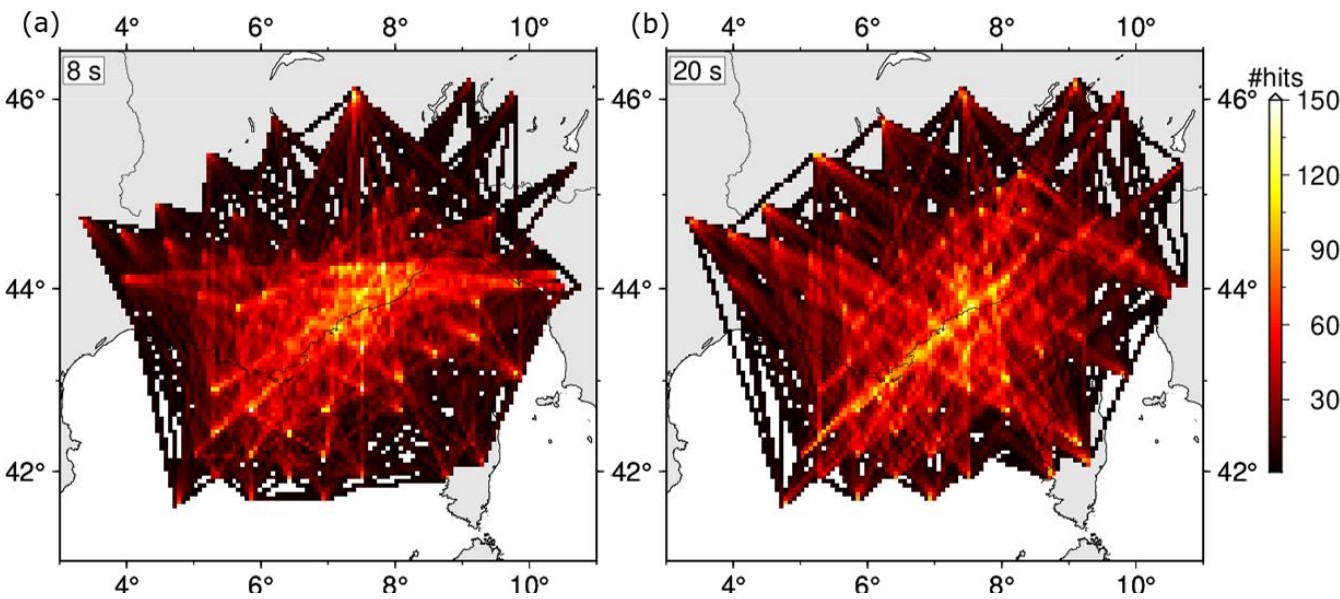

**Figure 2: Hit count maps showing the ray coverage for ambient noise CCF-pairs at 8 s (a) and teleseismic CCF pairs at 20 s (b). The**
**grid cells have a size of 5x5 km.**


*Group velocity dispersion curves - fundamental mode and higher modes*

To estimate group velocity dispersion curves, we apply the Multiple Filter Technique (MFT) (Dziewonski et al., 1969). A narrow bandpass filter is applied to the CCFs to derive the velocity for a distinct period from the maximum correlation (e.g. Meier et al., 2004).

Extra care has to be taken when picking the dispersion curves since for some station pairs the first higher mode has stronger amplitudes than the fundamental mode. Different modes have different sensitivity kernels (e.g. Harmon et al., 2007), and, unfortunately, our tomography program cannot process input data from more than one mode at a time. Therefore, we picked manually by comparing each ray path to the theoretical dispersion curves predicted from a 2D model of the research area (Fig. S2) that includes results from Makris et al. (1999), Gailler et al. (2009), and Dannowski et al. (2020).


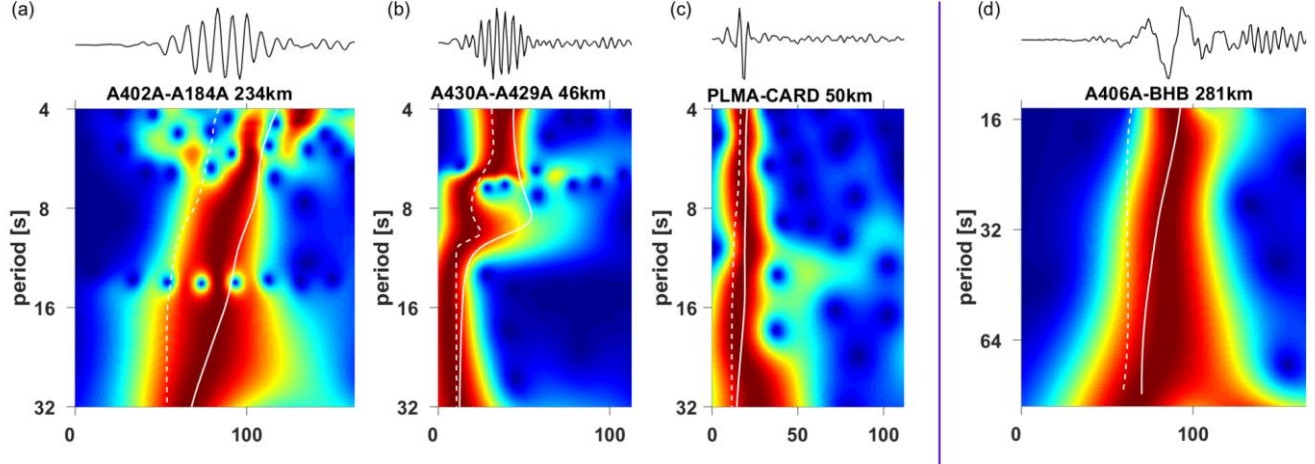

**Figure 3: MFT examples for correlations on (a) OBS-land pair, (b) OBS-OBS pair, (c) land-land pair (all ambient noise cross-correlations), and (d) land-land station pair (cross-correlation containing teleseismic event). The solid white line shows the theoretical fundamental mode; the dashed white line shows the theoretical first higher mode. In (a), (c), and (d), the theoretical**
**fundamental mode fits the theoretical velocities. In (b), the theoretical first higher mode correlates most strongly. Therefore, pair (b) was excluded from the tomography.**

First, we picked the maximum signal on all dispersion curves. Station pairs showing no detectable maximum were excluded. After comparing the group velocities with synthetic dispersion curves, we excluded about 100 station pairs that showed velocities likely related to higher modes (Fig. 3b). Higher modes were mainly observed for ray paths in the southern part of
the Ligurian Basin and parallel to the basin axis (Fig. 4). The origin could be layers in which the first higher mode couples more strongly than the fundamental mode, as previously observed by Takeo et al. (2014) for CCFs from OBS in the NW Pacific. During the MFT revision, we did not observe a degradation of the signal depending on the station distance.

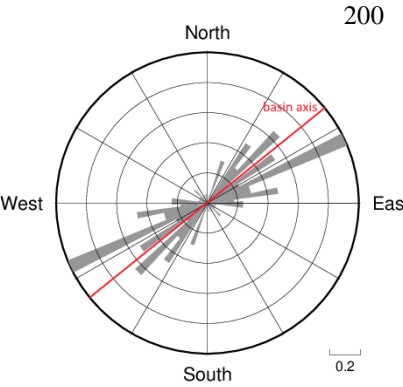


**Figure 4: Rose diagram of ray path azimuths showing the striking directions of the higher modes. The red line indicates the basin axis.**

Identifying and rejecting the higher mode dispersion curves resulted in 1342 dispersion curves for the fundamental mode from ambient noise CCFs and additional 1963 dispersion curves from teleseismic CCFs (Fig. 5) used for further analysis steps. We use the ambient noise CCFs to derive dispersion curves for periods from 4-15 s. The CCFs from the correlation of teleseismic events were used to derive dispersion curves from 20-90 s (Fig. 5). The dispersion curves' frequency bands are complementary and provide a bandwidth ranging from 4-90 s.

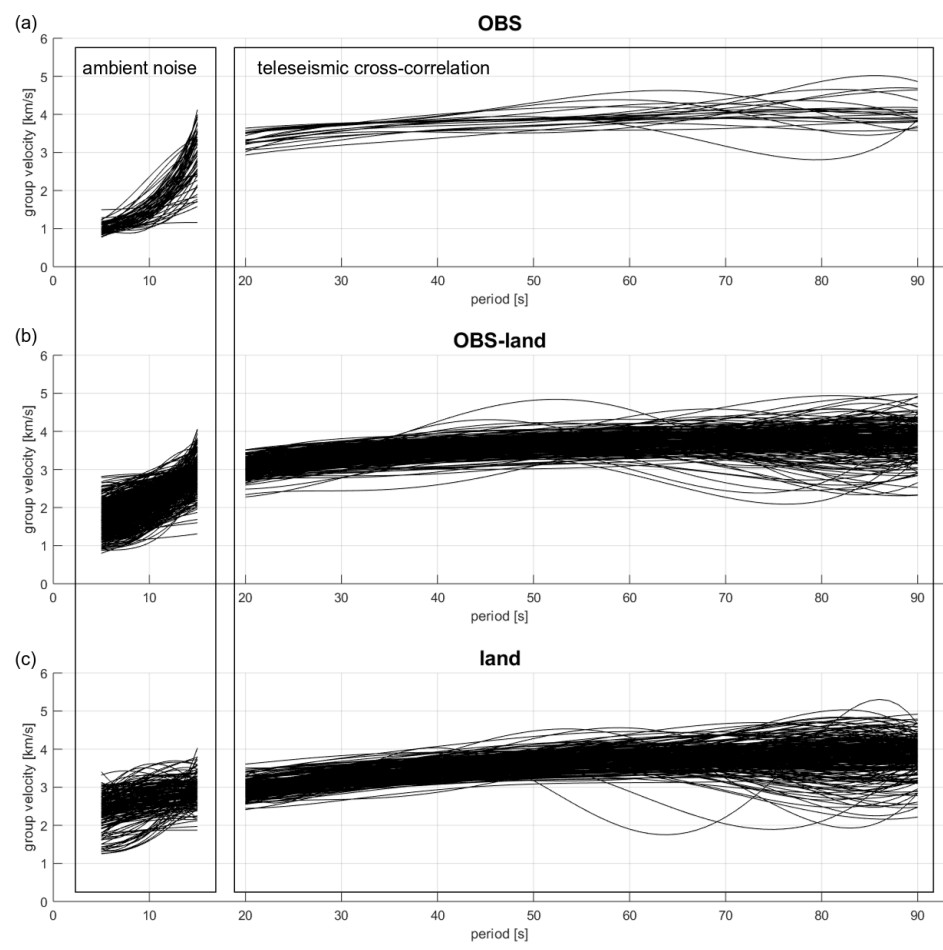

**Figure 5: Picked dispersion curves from ambient noise cross-correlation and correlation of teleseismic events. The dispersion curves are sorted for different types of station pairs: (a) OBS-OBS pairs, (b) OBS-land pairs and vice versa, (c) land-land pairs.**

### 3.3 Surface wave tomography for group velocities of ambient noise data and teleseismic data

We use the Fast Marching Surface Tomography method (FMST, Rawlinson & Sambridge, 2004, 2005) to derive 2D Rayleigh group velocity maps (Fig. 6 and Fig. S11) from the picked dispersion curves. FMST inverts for 2D map slices of group velocities for a given period. The forward prediction of travel times is achieved using the fast marching method (Sethian, 1996; Sethian and Popovici, 1999), a finite-difference solution of the eikonal equation. The inversion scheme is non-linear and repeated iteratively. Prior to the inversion, we deleted data outside the allowed velocity range: 0.5-3.6 km s$^{-1}$ for periods < 10 s; 1.0-4.0 km s$^{-1}$ for periods between 10-20 s; and 2.0-6.0 km s$^{-1}$ for periods 20 s and larger. We derived these thresholds based on the seismic velocity model of an active seismic refraction profile in the centre of the Ligurian Basin (Dannowski et al., 2020). The damping parameter for every period was estimated from L-curves (e.g. Hansen, 1992; Fig. S2). The smoothing


parameter was chosen visually depending on the resolution of the inversion (see Table S3 for inversion parameters) and the result of the checkerboard tests (Fig. 7). The input error is based on the picking error and linearly increases with the increasing

periods from ±0.75-2.0 s. We use homogeneous starting models with period-dependent velocities (Table S2). These are based on a group velocity model calculated from the seismic refraction line by Dannowski et al. (2020). The inversion grid consists of 28x35 nodes, resulting in one node every 18 km for both N-S and W-E directions. Due to grid refinements, the output grids consist of 406x511 grid points with a spacing of 0.0157°x0.0136° or 1.5x1.23 km. We calculate 2D group velocity maps for 5 s, 6 s, 7 s, 8 s, 9 s, 10 s, 12 s, and 15 s from ambient noise CCFs and 20 s to 90 s in 10 s steps from teleseismic CCFs.

In the initial group velocity maps, we observed a low-velocity area west of Marseille associated with station ARBF. We ran a tomography with all ray paths from ARBF excluded, and the result did not show the low-velocity area. Since the station is positioned on sediments in the outer Rhône delta, we assume the low-velocity zone to be caused by a locally 'slower' subsurface. We decided to exclude the station from our dataset to prevent the smearing of local low velocities into the group velocity maps. A similar low-velocity zone was observed close to station A430A, which was excluded as well.

Next, we use these initial group velocity maps to calculate residuals between the model input and the tomography output. We evaluate the residuals and keep those station pairs corresponding to 1.28 standard deviations (σ; 80 % of all pairs) for periods of 5 s to 15 s. For longer periods, we observe smaller residuals and therefore keep 90 % of the station pairs (1.64 σ). Then, we recalculate the 2D tomographies with the updated dataset (see Table S2 for final numbers) to create group velocity maps from ambient noise CCFs.


*1D depth inversion*

To remove effects of the highly variable topography and bathymetry, we invert for 1D shear-velocity-depth profiles using the iterative, weighted inversion code from Herrmann (2013). We produce one 1D-$v_S$-depth-profile for every 10th grid point, corresponding to one profile every 12.3 km. To account for the non-uniqueness of the solution (Foti et al., 2018), we set up a

starting model with fixed layers (Table S4) based on the $v_P$-velocities from Dannowski et al. (2020) for up to 16 km depth and on PREM (Dziewonski and Anderson, 1981) below. Since the dispersion curves for FMST represent a cumulated velocity profile for the subsurface between two stations, it is crucial to correctly parameterise the topography and water column prior to the velocity-depth inversions. To consider the effect of topography and bathymetry, we set up the two uppermost model layers independently: onshore, the top layer reaches from the local elevation to the sea level. The second layer reaches from

sea level to a depth of 4 km. Offshore, the uppermost layer represents the water column with fixed velocities of $v_P$=1.52 km s$^{-1}$ and $v_S$=0 km s$^{-1}$, reaching from the sea surface down to the seafloor, followed by a second layer below reaching from the seafloor down to 4 km depth. Therefore, below 4 km depth, all input models are identical. The layer thicknesses are not varied during the inversion, and the velocity uncertainty is estimated as 2 % of the input group velocity. After parameterisation, we perform iterative 1D depth inversions for $v_S$ (Herrmann, 2013). We obtain 1D velocity-depth profiles from the surface to a

depth of 30 km.

## 3.4 Data Quality

In general, the OBS stations have noise characteristics comparable to land data (Fig. S1). However, roughly 50 % of all possible CCFs combinations do not show a clear correlation of the group velocities and hence were not considered further. Each OBS is part of combinations resulting in high-quality and low-quality CCFs. Similar effects have been observed by Harmon et al. (2012) and Adimah & Padhy (2020). One reason for this may be the variability in station sites. For the OBS, the water depth is highly variable (1133 m to 2773 m). Also, the seafloor characteristics and the coupling to the subsurface are most likely very variable due to the varying sediment thickness beneath each site (Schettino and Turco, 2006). Overall, the essential difference between the AlpArray OBS stations and most previous studies (e.g. Harmon et al., 2007; Lin et al., 2016) is the shallow location of AlpArray OBS in the marginal Ligurian Basin. To our knowledge, this study provides the shallowest OBS water depths used for CCFs, and the shallow water might not prove beneficial for the correlation quality. Five stations are at water depths of 2 km or less (the shallowest station A434A is at 1.1 km depth), none is deeper than 2.8 km. Harmon et al. (2012) estimated CCFs of similar quality using OBS stations at 2.5-3.5 km water depth offshore Sumatra. Adimah & Padhy (2020) use OBS in deeper water (14 OBS in 4.3-5.1 km depth and only one OBS at 2.7 km). They observe variations in CCF quality as well, but their overall quality of CCFs is better than for our dataset.

Other reasons for our comparably low CCF quality include the form of the basin itself, for which noise sources are not uniformly distributed, and probably also the highly variable weather conditions in our research area. The Mediterranean Sea lies in a westerly wind system, but especially during winter, mistral events change the flow pattern of regional ocean currents (e.g. Millot & Wald, 1980; André et al., 2005). Moreover, mistral winds might create significant wave heights of 4 m and more (e.g. Pasi et al., 2011). Those temporary changes of the water column and currents alter the pressure on the OBS and the ocean floor and might therefore introduce highly variable noise. Additionally, the land station locations vary in topography and geological settings ranging from sediment basins to Alpine mountains. Nevertheless, overall we estimated more than 3300 high-quality dispersion curves.

*Resolution Tests*

To estimate the resolution of the group velocity maps, we calculated two checkerboard tests for every period (5 - 90 s) with tiles of 0.4°x0.4° and 0.8°x0.8°, respectively (Fig. 7 and Fig. S3-S6). The tiles' deviation from the input velocity was set to ± 25 %. Synthetic data are calculated and inverted, using the same setup as for the picked data. Overall, the resolution is good in the Ligurian Basin and along the northern coast. We defined one polygon where $v_G$ is reasonably well defined for all investigated periods. This was necessary to use the group velocity maps as input for the 1D shear-wave velocity inversion. Additionally, we performed a restoration test based on a synthetic 2D model of the research area (Fig. S7a). Different parts of the research area were assigned to distinct group velocity profiles (Fig. S7b). The synthetic group velocity maps for distinct periods are shown in Fig. S8. Based on this model, we calculated synthetic lagtimes for all station pairs used in the real dataset.

This was done by projecting the ray path and estimating a total lag time, based on the proportion that the ray travels through

the different areas. We then used this synthetic data set to calculate group velocity tomographies, using the same settings as for the real data set (Fig. S9). Comparing these to the synthetic group velocities (Fig. S8) supports our checkerboard test results. The Ligurian Basin itself and the Liguro-Provençal coast are well resolved. We observe some artefacts caused by the ray coverage (e.g. finger-like high-velocity areas in the north and from Corsica to the Italian mainland) that lie outside the interpreted area.

To evaluate the resolution of the 1D-shear-wave inversion, we used the above group velocity maps (Fig. S9) to run a synthetic 1D-shear-wave inversion based on the restoration test. For this, we also use the same setup as for the real data set. The resulting shear-velocity depth layers are a good hint of resolvable areas of the lithospheric structure (Fig. S10). Also, we estimate a root-mean-square (RMS) error for every 1D inversion.

**4 Results**

**4.1 Rayleigh wave group velocity**

We show 2D group velocity maps for periods of 5 s, 8 s, 12 s, 20 s, and 40 s (Fig. 6b-f) accompanied by the tomography input as coloured ray path plots for 8 s (Fig. 6a). The resolvable area, marked by the red polygon in Fig. 6b-f, is determined from the checkerboard tests (Fig. 7); poorly resolved parts are transparent in the group velocity maps. Group velocity maps for all

other periods used are shown in Fig. S11. The ray coverage for ambient noise tomography and the cross-correlation of teleseismic events differs (Fig. 2). Still, the resolved area of both data sets covers the Ligurian Basin and adjacent coastal areas (Fig. 7 and Fig. S3-S6), whereby the Liguro-Provençal coast is better resolved than the Corsican margin.

Along the Liguro-Provençal coastline, we observe a clear velocity change for periods of 5-12 s (Fig. 6b-d): $v_G \cong$ 1-1.5 km s$^{-1}$ offshore and $v_G \cong$ 2.5-3 km s$^{-1}$ onshore for 5 s and 8 s, $v_G \cong$ 2-2.5 km s$^{-1}$ offshore and $v_G \geq$ 2.8 km s$^{-1}$ onshore for 12 s. For

longer periods, this distinction becomes less sharp, and the velocity gradient changes direction. For 20 s and 40 s (Fig. 6 e,f), $v_G$ is approximately 0.5 km s$^{-1}$ slower onshore compared to the Ligurian Basin. The group velocity maps for periods 20 s and 40 s appear more homogenous than for shorter periods. For 20 s we observe $v_G$ = 3-3.5 km s$^{-1}$, for 40 s it is $v_G$ = 3.5-4 km s$^{-1}$. The Ligurian Basin appears to be separated into a southwestern (labelled SW in Fig. 6c) and a northeastern part (labelled NE in Fig. 6c) of the Ligurian Basin. The onshore-offshore separation appears less distinct in the northeastern basin, where the

group velocity increases gradually towards the coast. In short periods, the NE part of the basin is faster (NE: 1.5-2.5 km s$^{-1}$ at 5 s, SW: ~1 km s$^{-1}$) than the southwestern part. At 12 s, the velocity gradient is smaller (NE: ~2.5 km s$^{-1}$, SW: ~2 km s$^{-1}$), and at 20 s the gradient vanishes.

Overall, the group velocity increases with increasing period. The velocity gradient is strongest (5 s period: $v_G \cong$ 1km s$^{-1}$; 12 s period: $v_G$ = 2-3 km s$^{-1}$) beneath the southwestern basin, less strong ($v_G \cong$ 1.75 km s$^{-1}$ to 2.5 km s$^{-1}$) beneath the northeastern

basin and least strong ($v_G \cong$ 2.5 km s$^{-1}$ to 3-3.25 km s$^{-1}$) beneath the mainland.

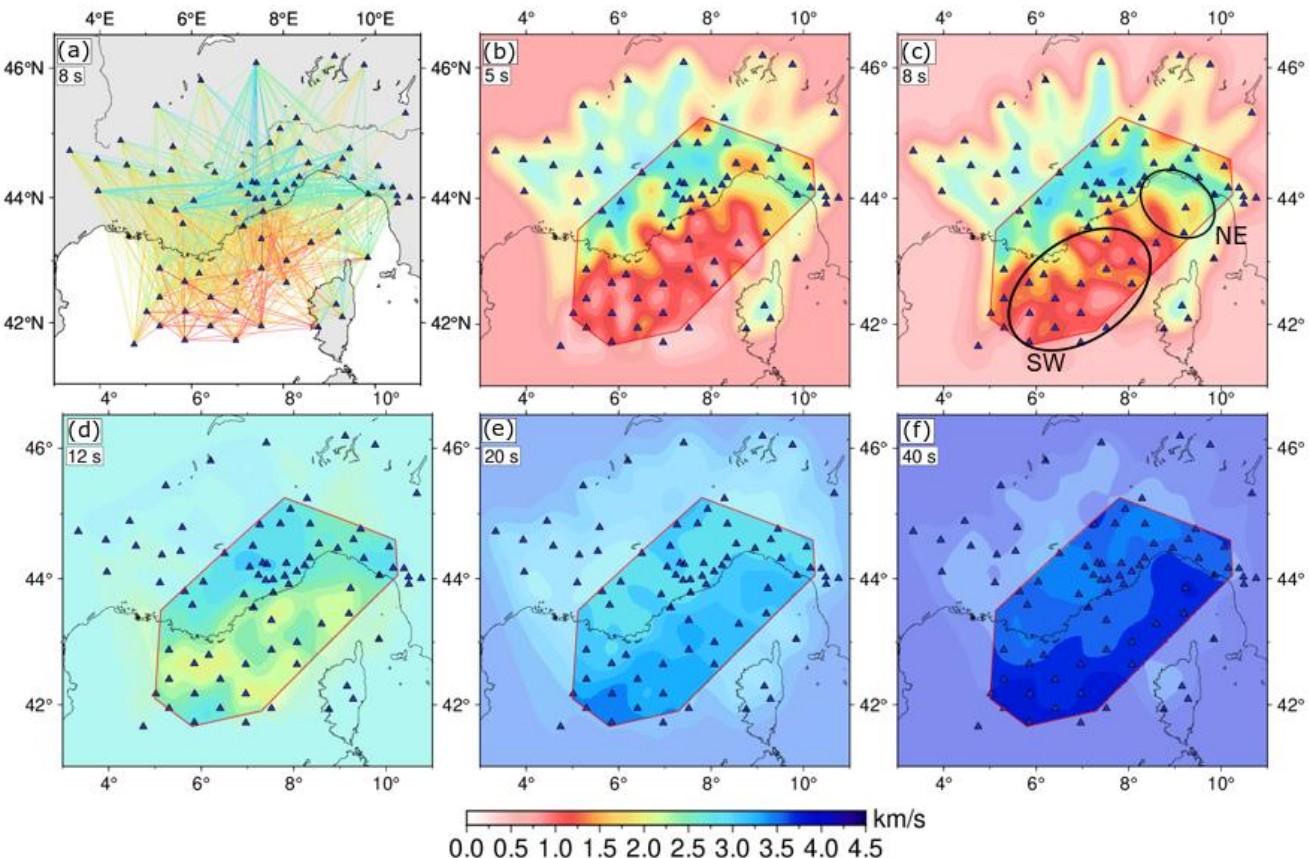

**Figure 6: Panel (a) shows the tomography input for 8 s as a ray path plot. Panels (b)-(f) show group velocity maps of the Ligurian Basin from surface wave tomography for 5 s, 8 s, 12 s, 20 s, and 40 s period, whereby (b), (c), and (d) are based on ambient noise cross-correlation and (e) and (f) are based on the cross-correlation of teleseismic events. A red polygon marks the resolved area. Areas of low resolution are shown in transparent colours; areas without ray coverage show the initial velocity. Annotations in (c) mark the southwestern and central (SW) and the northeastern (NE) Ligurian Basin. Blue triangles represent stations.**


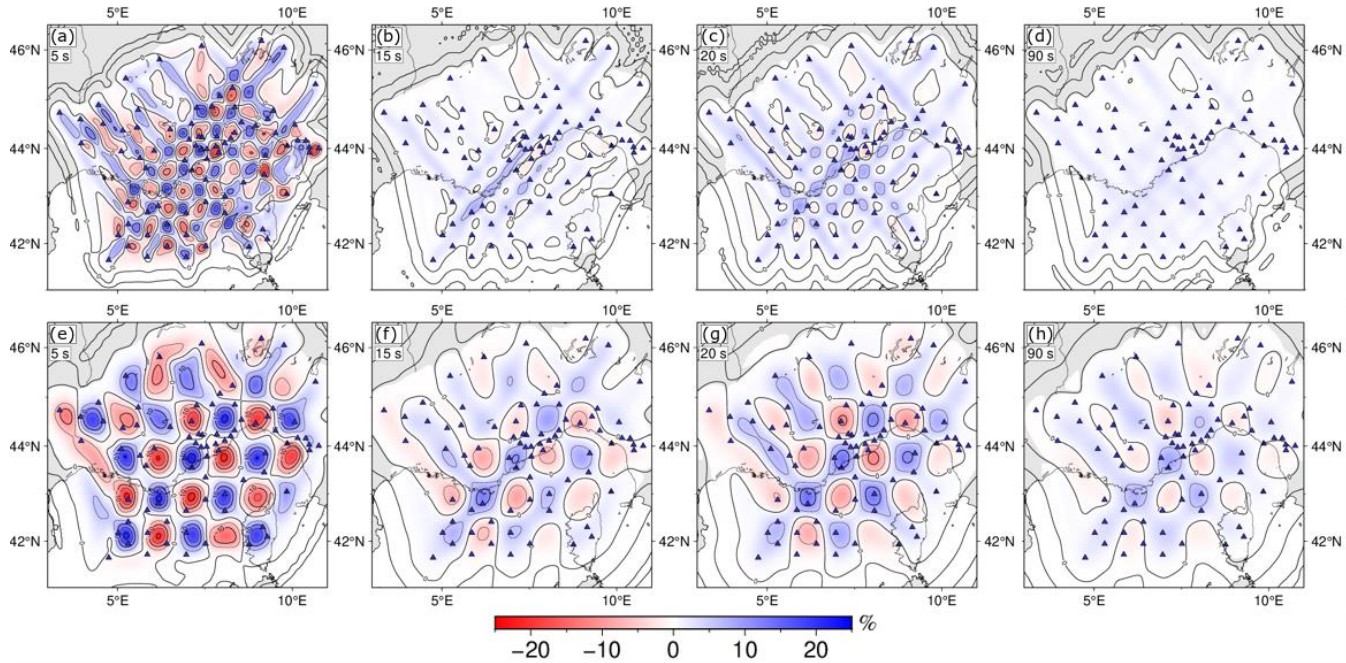


**Figure 7: Checkerboard tests for periods of 5 s, 15 s (both ambient noise cross-correlation), 20 s, and 90 s (both teleseismic cross-correlation). The perturbation of the input checkerboard tiles is set to ±25 %. Panels (a)-(d) show checkerboard tests with a grid size of 0.4°x04°, panels (e)-(h) show a grid size of 0.8°x0.8°. The standard deviation of Gaussian noise (of travel times) is set to 0.375 s for (a) and (e), 0.665 s for (b) and (f), 0.708 s for (c) and (g), and 1 s for (d) and (h). Checkerboard tests for all other periods can be**
**found in the Supplement (Fig. S3-S6). Blue triangles represent stations.**

### 4.2 1D shear-wave velocity inversion

We calculated 1D depth inversions for $v_S$ based on the group velocity maps (5-90 s) described above. It was a crucial step to remove the topographical effects that result from the amphibious nature of our study area. The average RMS of the 1D
inversions is 0.15 km s$^{-1}$ (Fig., 8i).

*Liguro-Provençal coast*

At shallow depth, the velocity structure onshore is heterogeneous. At a depth of 6-9 km below sea level (Fig. 8c), we see $v_S \cong$ 2.75-3 km s$^{-1}$ for the Po plain, $v_S \geq 3.5$ km s$^{-1}$ along the Alpine belt, $v_S \cong$ 2.7-3 km s$^{-1}$ west of Nice, lower $v_S$ directly at the
coast, and an increase in $v_S$ ($v_S \cong 3$ km s$^{-1}$) towards the Maures-Esterel Massif (Fig. 1). Just onshore Liguria (Fig. 8c), our results also indicate a narrow band of $v_S \cong 3.5$ km s$^{-1}$ in 6-9 km depth accompanied by lower $v_S \cong 3.2$ km s$^{-1}$ offshore.

At 9-12 km depth, we observe a high-velocity area north of Nice (dashed circle in Fig. 8d), showing $v_S \cong 4.2$ km s$^{-1}$. In other depth layers, this area does not show large velocity differences compared to the surrounding area. In up to 12-15 km depths, we observe high S-wave velocities beneath the Alpine belt that decrease towards the Rhône delta in the southwest and towards
the Po plain in the northeast (Fig. 8a-e). At larger depths, the high-velocity anomaly disappears, and the velocity field along

the coastline gets smoother. At 18-21 km depth (Fig. 8g), we observe large areas of $v_S = 3.5$ km s$^{-1}$. In 21-25 km depth (Fig. 8h), the velocity reaches $v_S = 4$ km s$^{-1}$ locally.

*Southwestern and central Ligurian Basin*

In the southwestern and central basin (labelled SW in Fig. 8a), the shear-wave velocities in the uppermost 4 km (Fig. 8a) are ~1.5 km s$^{-1}$. The velocity increases towards the Provençal coast and the Gulf of Lion. The 4-6 km layer (Fig. 8b) shows $v_S \cong$ 2 km s$^{-1}$ with areas of higher velocity ($v_S \geq 3$ km s$^{-1}$) offshore Marseille and northwest of Corsica. Throughout the basin, but mainly along the basin axis, we observe areas of higher S-wave velocity of up to 3.5 km s$^{-1}$ in 6-9 km depth (Fig. 8c). With increasing depth, the S-wave velocity increases and in 12 km depth, $v_S \geq 4.3$ km s$^{-1}$ is reached locally in the basin centre.

These fast areas broaden in the 12-15 km depth slice (Fig. 8e). The S-wave velocity is slower towards the Provençal coast. At 15-18 km depth (Fig. 8f), we observe $v_S \geq 4.3$ km s$^{-1}$ along the basin axis of the whole southwestern and central Ligurian Basin. However, $v_S$ is slower (3.7-4 km s$^{-1}$) south of Marseille and in the outer Gulf of Lion. At a depth of approximately 21 km (Fig. 8g), $v_S \geq 4.3$ km s$^{-1}$ applies to most of the southwestern and central basin, except for the aforementioned areas. Close to the Provençal coast, $v_S \geq 4.3$ km s$^{-1}$ is reached only in the 21-25 km depth layer (Fig. 8h).

The "fingers" of high $v_S$ leading from the basin axis towards the coast east and west of Nice (e.g. Fig. 8g) are probably caused by an insufficient ray coverage of the group velocity tomography in that area. The ray coverage is better offshore Nice. Therefore, we expect a similar $v_S$ as offshore Nice ($v_S = 3.5$ km s$^{-1}$).

*Northeastern basin*

In the northeastern basin (labelled NE in Fig. 8a), the shear-wave velocity is higher than in the southwestern basin for shallow depths. North of Corsica, $v_S \cong 2$ km s$^{-1}$ in up to 4 km depth (Fig. 8a) with higher velocity $v_S \cong 2.5$ km s$^{-1}$ close to the Italian coast. The offshore velocity increases to $v_S \cong 2.5$-3 km s$^{-1}$ at 4-6 km depth (Fig. 8b) and $v_S \cong 3$ km s$^{-1}$ at 6-9 km below the sea surface (Fig. 8c). In both layers, we identify an area of higher velocity northeast of Corsica. This patch shows $v_S > 3$ km s$^{-1}$ in the 4-6 km layer (Fig. 8b) and $v_S > 3.5$ km s$^{-1}$ in 6-9 km depth (Fig. 8c). From 9 km to up to 21 km depth (Fig. 8d-g), the

offshore $v_S$ increases slowly from approximately 3.5 km s$^{-1}$ to 3.8 km s$^{-1}$. Close to the Italian coast, the velocity gradient direction switches at approximately 12-15 km depth (Fig. 8e). For deeper layers, $v_S$ is lower near the Italian coast than towards the southeastern basin. At 21-25 km depth (Fig. 8h), $v_S \geq 4.3$ km s$^{-1}$ accounts for the whole basin, except for a narrow band at the Ligurian coast that shows lower velocities of $v_S \cong 4$ km s$^{-1}$.

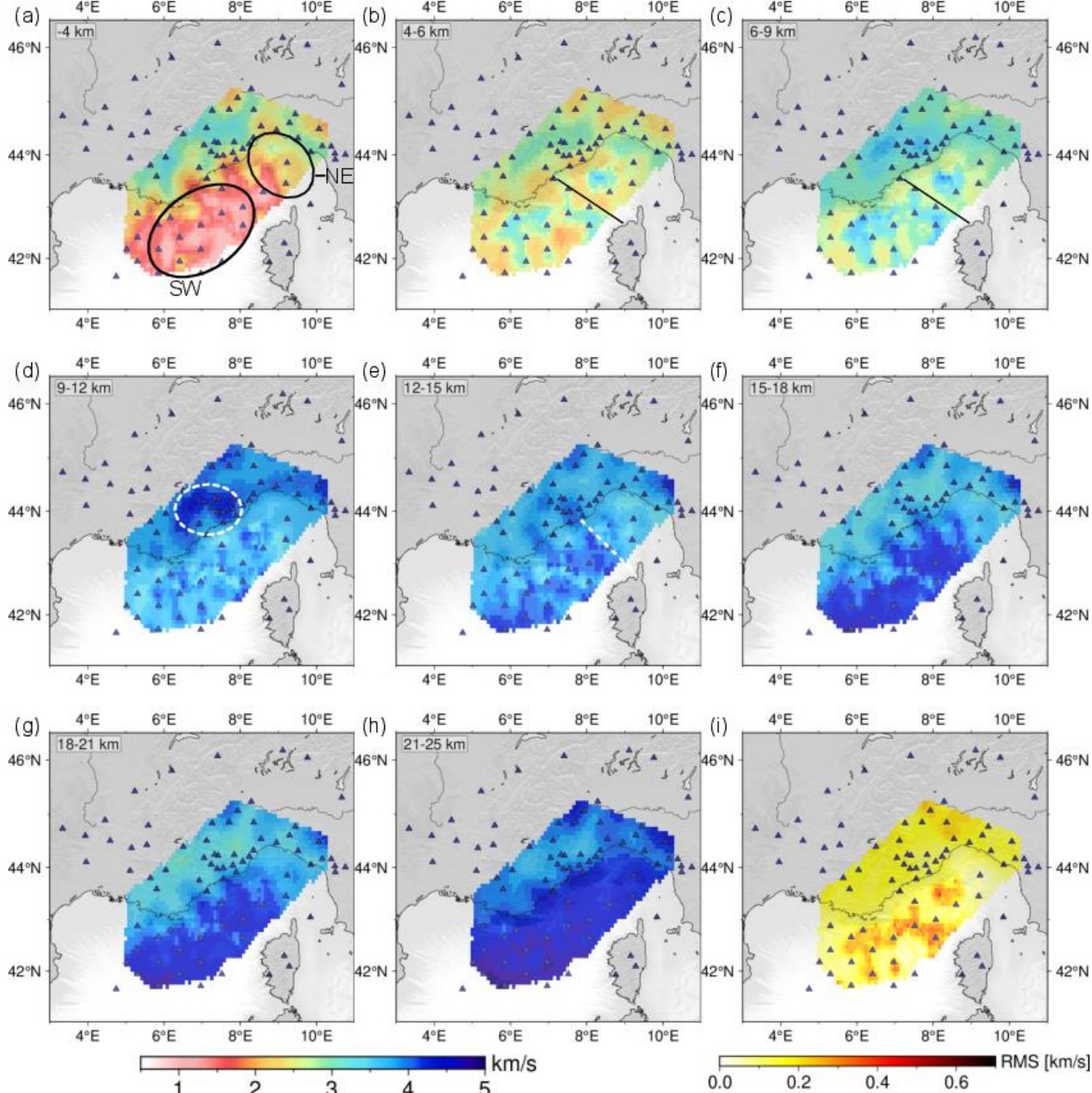

**Figure 8: 2D shear velocity maps derived from the 1D inversion. Layer depth is stated in the upper left corner. Depths (in km) are relative to the sea surface. The annotations in (a) mark the southwestern and central (SW) and the northeastern (NE) Ligurian Basin. The solid black line in (b) and (c) show the location of profile LISA01 (Contrucci et al., 2001). The dashed circle in (d) marks a high-velocity area north of Nice (see Sect. 4.2.1), and the dashed white line in (e) represents the proposed prolongation of the Alpine front (Rollet et al., 2002). Layer 1 (topography), layer 10 (25-30 km), and layer 11 (halfspace) are shown in Fig. S3. Panel (i) shows**

 **the root mean square (RMS) value for the 1D shear-wave-inversion in mapview (i.e. one RMS value per grid point). Blue triangles mark stations.**

## 5 Discussion and Geological interpretation

In the following, we discuss three regions that show differing characteristics in the velocity maps: the Liguro-Provençal coast, the southwestern and central Ligurian Basin, and the northeastern basin. Also, we discuss the proposed offshore prolongation of the Alpine front.

### 5.1 Liguro-Provençal coast

Along the Liguro-Provençal coast, we can compare our results to existing larger-scale ambient noise studies from Molinari et al. (2015b) and Kästle et al. (2018), as well as a local ambient noise study by Guerin et al. (2020). Guerin et al. (2020) conducted an ambient noise study covering the Provençal coast from Marseille to the Argentera Massif north of Nice (Fig. 1). Guerin et al. (2020) show Rayleigh wave group velocities as coloured ray coverage maps, as we do for 8 s period (Fig. 6a). At 8 s period, we observe $v_G$=2.75-3.2 km s$^{-1}$ in the coastal area. Guerin et al. (2020) find approximately $v_G$ = 3 km s$^{-1}$ (their Fig 6). In shear-wave velocity maps, Guerin et al. (2020) observe $v_S \cong$ 3-3.5 km s$^{-1}$ at a depth of 6.4 km (their Fig. 12) along the Provencal coast. This fits our results nicely (Fig. 8c). For shallower depths, Guerin et al. (2020) found that the S-wave velocity increases with depth faster than in our data set, a feature that is probably controlled by a denser station spacing compared to our study. At up to 9 km depth (Fig. 8a-c), we observe laterally varying shear-wave velocities on land that we assume to be caused by variations in the geology. At the Rhône delta (Fig. 1), where the sedimentary cover is up to 12 km thick (Pichon et al., 2010), we observe $v_S \cong$ 2.7 km s$^{-1}$ in the layer at 4-6 km depth (Fig. 8b) and $v_S \cong$ 3 km s$^{-1}$ in 6-9 km depth range (Fig. 8c). Similarly, the Po Basin has an average sedimentary thickness of 4-5 km (Molinari et al., 2015a) with a shear-wave velocity increasing from $v_S \cong$ 2.5 km s$^{-1}$ to $v_S \cong$ 3.1 km s$^{-1}$ at 4-9 km depth. In contrast to the sedimentary basins, we observe higher $v_S \cong$ 3-3.5 km s$^{-1}$ (4-9 km depth) beneath the Alpine belt, composed of crystalline and metamorphic rocks (e.g. Molinari et al., 2015b). This S-wave variation is most probably caused by the different rock types and structure of the Alpine belt and the sedimentary basins. West of Nice, we observe $v_S \cong$ 2.7-3 km s$^{-1}$ in 6-9 km depth (Fig. 8c), lower $v_S$ directly at the coast, and an increase in $v_S$ ($v_S \cong$ 3 km s$^{-1}$) towards the Maures-Esterel Massif (Fig. 1). These results all compare well to Molinari et al. (2015b, compare their Fig. 8). The large-scale structures also compare nicely to the results of Kästle et al. (2018, their Fig. 9) and Lu et al. (2018, their Fig. 7). In contrast to Molinari et al. (2015b), they observe slightly higher velocities in 10 km depth that are closer to those in our 9-12 km depth slice (Fig. 8d). Both our results and Kästle et al. (2018) indicate a narrow band of $v_S \cong$ 3.5 km s$^{-1}$ in 6-9 km depth just onshore Liguria (Fig. 8c), accompanied by lower $v_S \cong$ 3.2 km s$^{-1}$ offshore. This observation of a local

high-velocity area onshore Liguria is supported by seismic refraction profiles evaluated by Laubscher et al. (1992). They identified several high-velocity bodies ($v_P > 6$ km s$^{-1}$) linked to ophiolites.

Our results show a high-velocity area ($v_S \cong 4.2$ km s$^{-1}$) north of Nice at 9-12 km depth (dashed circle in Fig. 8d), coinciding with a small area of higher velocity in the 10 km depth map of Kästle et al. (2018). This is probably linked to the Argentera Massif (Fig. 1). The Massif is composed of crystalline rocks and was identified as a high-velocity area in more shallow depths

($v_S \cong 3.4$ km s$^{-1}$ at 6 km depth) by Guerin et al. (2020). The high-velocity anomaly cannot be tracked in greater depth (Fig. 8e). Instead, the velocity field along the coastline gets smoother. At 18-21 km depth (Fig. 8g), we observe large areas of $v_S = 3.5$ km s$^{-1}$. This velocity is similar to Molinari et al. (2015b) and Kästle et al. (2018). We still observe crustal velocities ($v_S \cong 4$ km s$^{-1}$) in 21-25 km depth (Fig. 8h), in line with the Moho depth of ~35 km beneath the Liguro-Provençal coast (Kästle et al., 2018).

**5.2 Southwestern and central Ligurian Basin**

At shallow depths (Fig. 8a-b), the S-wave velocity is mainly dominated by sediments. We observe $v_S = 1$-1.5 km s$^{-1}$ in up to 4 km depth (below the sea surface) and $v_S = 2$-2.5 km s$^{-1}$ in 4-6 km depth. Studies by Schettino & Turco (2006) revealed thick sediment layers in the southwestern Ligurian Basin. Offshore western Corsica, the sediments are 6-7 km thick, with the maximum thickness of 8 km occurring to the southwest of Marseille (e.g. Schettino and Turco, 2006). This is supported by the

findings of Moulin et al. (2015). Their wide-angle reflection seismic data show up to 7.6 km of sediment in the southeastern Gulf of Lion, thinning to 6.3 km in the Ligurian Basin. Throughout the basin, but mainly along the basin axis, we observe areas of higher S-wave velocity of up to 3.5 km s$^{-1}$ in 6-9 km depth (Fig. 8c). These are also the areas with the highest RMS error (Fig. 8i). Some of these, e.g. north of Corsica, are in locations where Rollet et al. (2002) observe magmatic anomalies related to magmatic intrusions. We deduce that the velocity gradient is stronger for fast areas along the basin axis than away

from the basin axis. The changing gradient is probably caused by the observed thinning of continental crust (Dannowski et al., 2020) and possible exhumation of denser lower crust and upper mantle rock (Gailler et al., 2009; Jolivet et al., 2015) observed further to the southwest. Both would lead to a higher S-wave velocity near the basin axis.

Dannowski et al. (2020) observe a Moho depth of about 12 km in the basin centre, where we observe patches of $v_S \geq 4.3$ km s$^{-1}$ along the basin axis (Fig. 8e). Comparing the P-wave velocity of Dannowski et al. (2020) to our S-wave velocity, we

calculate a $v_P/v_S$-ratio of $7.5/4.3 = 1.74$ at the southwestern end of their profile. Following Carlson & Miller (2003), this does not indicate mantle serpentinisation. This interpretation is also supported by the local seismicity study of Thorwart et al. (2021). They observe $v_P = 8.1$ km s$^{-1}$ and $v_S = 4.7$ km s$^{-1}$ ($v_P/v_S = 1.72$) in the basin centre roughly 3-4 km below the Moho. The fast area along the axis broadens in the 12-15 km depth slice (Fig. 8f). Linking this observation to the seismic lines (e.g. Jolivet et al., 2015; and Dannowski et al., 2020), our results indicate that the Moho depth increases towards the Provençal and Corsican

margins. At larger depths, the velocity maps get more homogeneous. This hints at fewer heterogeneities in the mantle but might also be caused by the decreasing sensitivity of group velocities with increasing periods (e.g. Adimah & Padhy, 2020).

For a 5 s period, the group velocity is sensitive to a narrow depth range that peaks at ~5 km. For 20 s, the overall sensitivity is lower and has a much broader range of approximately 10-25 km depth.

In the central Ligurian Basin, Contrucci et al. (2001) investigated a multichannel seismic profile (LISA01) from Antibes, close to Nice, to L'Île Rousse on Corsica (Fig. 1 and Fig. 8b-c). They find that the transition from sediments to crust (at $v_P$ ~ 4.8-5 km s$^{-1}$) is shallow at the Provençal coast (3 km below sea level), deepens towards the basin centre (8 km below sea level), and rapidly shallows again at the Corsican margin (from 5 km to 1.5 km below sea level). Also, the salt (Messinian) and sediment (Miocene) layers ($v_P = 3.8$ km s$^{-1}$ to 5 km s$^{-1}$) thicken towards the basin centre (Contrucci et al., 2001), where the Moho is ~12-
13 km deep. Our $v_S$ maps for 4-9 km depth (Fig. 8b-c) show a local velocity high with increasing $v_S$ (2.75-3 km s$^{-1}$) directly offshore Nice. Further southeast along the LISA01 profile (Fig. 8b-c, solid black line), the velocity decreases to $v_S \cong 2.1$ km s$^{-1}$ in 4-6 km depth and $v_S \cong 2.5$-2.7 km s$^{-1}$ in 6-9 km depth. The resolution is poor at the Corsican margin, but $v_S$ increases to 3 km s$^{-1}$ towards Corsica (Fig. 8c). The observed velocity structure fits the findings of Contrucci et al. (2001) nicely, supporting their finding of thicker sediment and salt layers near the basin axis. Comparing $v_P$ of the LISA01 profile to our $v_S$ gives a $v_P/v_S$-
ratio of 1.75 for the high-velocity area offshore Nice and $v_P/v_S \cong 2.1$ for the sediment layers at the basin axis. Shillington et al. (2007) found similar values for sediments up to 1 km below the seafloor.

Dannowski et al. (2020) suggest that continental crust was (extremely) thinned along their profile, but that no spreading occurred. This is in-line with our results. A possible spreading centre must be located to the southwest. At the Gulf of Lion margin along the southwestern edge of our research area, Gailler et al. (2009) interpreted their results as oceanic crust, also
observing a transition zone made up of "lower crustal material or mixture of serpentinised upper mantle material with lower crustal material" (Gailler et al., 2009). Later, Jolivet et al. (2015) attributed the shallow high velocities to exhumed lower crust and possibly also partially serpentinised mantle. Therefore, a spreading center may have been located southwest of our research area, possibly as close as the Gulf of Lion margin.

**5.3 Northeastern basin**

Northeast of the LISA01 profile, the northeastern basin (labelled NE in Fig. 8a) exhibits different characteristics than the southwestern and central Ligurian Basin. For shallow depths of up to 12-15 km, the S-wave velocity is higher in the northeastern basin, compared to the southwestern basin. For greater depth, this switches and the northeast is slower. The velocity increase with depth is smaller in the northeast compared to the southwest. Overall, the northeastern basin is more
homogenous than the southwest, and the transition from the basin to onshore Italy is not as sharp as at the Provençal coastline. These observations are supported by large-scale ambient noise studies by Molinari et al. (2015b) and Kästle et al. (2018), observing a similar velocity distribution.

The sediment thickness map by Schettino & Turco (2006) shows a sediment thickness of 3-4 km in the northeast, increasing to the southwest. The northeastward thinning of the sediment layer explains the higher $v_S$ at shallow depths (Fig. 8b), compared

 to the southwest. Additionally, Makris et al. (1999) suggest that the sediments' compactness increases to the northeast. Increasing compactness would add to the velocity increase.

The crust-mantle boundary is well defined along seismic profiles across the Ligurian Basin. Our shear-velocity model adds spatial information to these studies, allowing for a broader understanding of the Moho. For the northeastern basin, we observe $v_S < 4$ km s$^{-1}$ in the 18-21 km layer and mantle-like $v_S \geq 4.3$ km s$^{-1}$ in the 21-25 km depth layer. This compares well to the Moho depth of 22 km near the Italian coast, observed by (Dannowski et al., 2020). They observe an increasing Moho depth towards the northeast. Linking our 3D shear-wave velocity data to the seismic observations indicates that the Moho depth gradually increases towards the northeast and from the basin axis towards the Provençal coast. Close to the Ligurian coast, we observe mantle-like $v_S \geq 4.3$ km s$^{-1}$ in most of the 21-25 km layer, except for a slim band of lower $v_S$ at the coast. At the coastline, Kästle et al. (2018) predict a Moho depth of 30-40 km. The apparent thickening of the continental crust towards the northeast is likely related to the position of the rotational pole of the opening of the Ligurian Basin during the Oligocene-Miocene. According to Speranza et al. (2002) and Gattacceca et al. (2007), the rotational pole was located in the northeastern Ligurian Basin at 43.5°N, 9.5°E. Therefore, the southwestern basin was more extensively opened, and the continental crust was thinned further than in the northeast.

**5.4 Alpine front**

Rollet et al. (2002) raised the question of whether an offshore prolongation of the Alpine front can be observed onshore France and onshore Corsica. These authors suggested that the southwestern and northeastern parts of the Ligurian Basin form, respectively, the footwall and hanging wall of the Alpine front. Thus, the Alpine front would be located approximately at the boundary between the northeastern and southwestern crustal domains distinguished in our data (illustrated by the dashed line in Fig. 8e). Dannowski et al. (2020) observe a gradual thickening of the continental crust towards the northeastern part of the Ligurian Basin. To explain the free-air anomaly derived by Sandwell et al. (2014), they did not need the sharp step that Makris et al. (1999) introduced between Corsica and the Liguro-Provencal coast. In keeping with Dannowski et al. (2020), our spatial shear-wave velocity data does not show a sharp lateral boundary, but a gradual change of the velocity layers. Detection of an offshore Alpine Front is therefore not feasible with the current resolution.

**6 Conclusions**

Applying ambient noise techniques and the correlation of teleseismic events to amphibious data results in the first 3D high-resolution seismic group and shear velocity models for the Ligurian Basin. Data processing of the OBS data included correction for tilt and compliance. The dataset differs from most previous ambient noise studies using OBS data. Our stations are comparably shallow, and the fundamental mode is not always the most prominent signal in the marine ray paths. Higher modes

are primarily observed in the southeast. Onshore, our results compare well with existing larger-scale ambient noise studies. We reveal a high-velocity area at the Argentera Massif, approximately 10 km below sea level. Offshore, the lithospheric structure in the Ligurian Basin mostly mimics the geometry of the basin. Shear-wave velocity maps indicate a gradual deepening of the Moho from 12-15 km in the southwestern basin centre towards 20-25 km in the northeastern basin and a more rapid deepening from the basin axis to the Provençal coast (> 30 km). Based on the low $v_P/v_S$ ratios of 1.74, we exclude mantle serpentinisation in the basin centre. Overall, the off-shore region north of Corsica is faster than the southwestern basin at shallow depths (<12 km) and slower at greater depth. This is linked to the varying sediment cover and the crustal thickness. In the southwestern part, the opening of the basin is more developed, but we do not observe oceanic crust in our study area. The change between these domains appears gradual.

**Data availability**

The data can be accessed via GEOFON and EIDA Data Archives. Data from AlpArray stations (including the OBSs) are accessible to AlpArray members. They will be freely accessible after March 2022.
We provide a zip-folder ('SE-wolfetal-2021-datasupplement.zip') as supplementary material. It contains the lagtime-input for the 2D group velocity tomography, the resulting grids, the codes we use to create the 1D inversion input (from group velocity maps) and the shear-wave velocity *.xyz files resulting from the 1D shear-wave inversion.

**Team list**

Complete list of the AlpArray Working Group:
György HETÉNYI, Rafael ABREU, Ivo ALLEGRETTI, Maria-Theresia APOLONER, Coralie AUBERT, Simon BESANÇON, Maxime BÈS DE BERC, Götz BOKELMANN, Didier BRUNEL, Marco CAPELLO, Martina ČARMAN, Adriano CAVALIERE, Jérôme CHÈZE, Claudio CHIARABBA, John CLINTON, Glenn COUGOULAT, Wayne C. CRAWFORD, Luigia CRISTIANO, Tibor CZIFRA, Ezio D'ALEMA, Stefania DANESI, Romuald DANIEL, Anke DANNOWSKI, Iva DASOVIĆ, Anne DESCHAMPS, Jean-Xavier DESSA, Cécile DOUBRE, Sven EGDORF, ETHZ-SED Electronics Lab, Tomislav FIKET, Kasper FISCHER, Wolfgang FRIEDERICH, Florian FUCHS, Sigward FUNKE, Domenico GIARDINI, Aladino GOVONI, Zoltán GRÁCZER, Gidera GRÖSCHL, Stefan HEIMERS, Ben HEIT, Davorka HERAK, Marijan HERAK, Johann HUBER, Dejan JARIĆ, Petr JEDLIČKA, Yan JIA, Hélène JUND, Edi KISSLING, Stefan KLINGEN, Bernhard KLOTZ, Petr KOLÍNSKÝ, Heidrun KOPP, Michael KORN, Josef KOTEK, Lothar KÜHNE, Krešo KUK, Dietrich LANGE, Jürgen LOOS, Sara LOVATI, Deny MALENGROS, Lucia MARGHERITI, Christophe MARON, Xavier MARTIN, Marco MASSA, Francesco MAZZARINI, Thomas MEIER, Laurent MÉTRAL, Irene MOLINARI, Milena MORETTI, Anna NARDI, Jurij PAHOR, Anne PAUL, Catherine PÉQUEGNAT, Daniel PETERSEN, Damiano PESARESI,

Davide PICCININI, Claudia PIROMALLO, Thomas PLENEFISCH, Jaroslava PLOMEROVÁ, Silvia PONDRELLI, Snježan PREVOLNIK, Roman RACINE, Marc RÉGNIER, Miriam REISS, Joachim RITTER, Georg RÜMPKER, Simone SALIMBENI, Marco SANTULIN, Werner SCHERER, Sven SCHIPPKUS, Detlef SCHULTE-KORTNACK, Vesna ŠIPKA, Stefano SOLARINO, Daniele SPALLAROSSA, Kathrin SPIEKER, Josip STIPČEVIĆ, Angelo STROLLO, Bálint SÜLE, Gyöngyvér SZANYI, Eszter SZŰCS, Christine THOMAS, Martin THORWART, Frederik TILMANN, Stefan UEDING, Massimiliano VALLOCCHIA, Luděk VECSEY, René VOIGT, Joachim WASSERMANN, Zoltán WÉBER, Christian WEIDLE, Viktor WESZTERGOM, Gauthier WEYLAND, Stefan WIEMER, Felix Noah WOLF, David WOLYNIEC, Thomas ZIEKE, Mladen ŽIVČIĆ, Helena ŽLEBČÍKOVÁ

**Author contribution**

Dietrich Lange, Martin Thorwart, Ingo Grevemeyer and Heidrun Kopp were responsible for the conception of this study. Dietrich Lange, Wayne Crawford, Anke Dannowski, and Heidrun Kopp were responsible for designing the OBS network. Felix Noah Wolf, Dietrich Lange, Anke Dannowski, Martin Thorwart, Wayne Crawford, and Heidrun Kopp acquired OBS data. Felix Noah Wolf analysed the data with the support of all co-authors. Wayne Crawford and Lars Wiesenberg provided software and expertise for the compliance correction and multiple filter technique. All authors interpreted the data. Felix Noah Wolf prepared the manuscript, and all authors critically reviewed it.

**Competing interests**

The authors declare that they have no conflict of interest.

**Acknowledgements**

This contribution is part of the German priority program "Mountain Building Processes in Four Dimensions (MB-4D) "SPP 2017 (www.spp-mountainbuilding.de/index.html) and of the international research initiative AlpArray. It is funded by the German research foundation (DFG) under grant-number LA 2970/3-1. We thank the captains and crews of RV *Pourquoi Pas?* and RV *Maria S. Merian* for their effort during deployment and recovery of the OBS network. We also thank the participating scientific crews. The *Deutscher Geräte-Pool für amphibische Seismologie* (DEPAS) provided twelve instruments; the *Institut de physique du globe de Paris* provided seven instruments. We thank Anne Paul for attending the cruise and her support with the OBS from the *Institut de physique du globe de Paris*. The deployment of the French component of the AlpArray Seismic Network was funded by the AlpArray-FR project of the Agence Nationale de la Recherche (contract ANR-15-CE31-0015). In addition to the AlpArray OBS array data, we used temporary and permanent land-based stations from the following networks: AlpArray (http://data.datacite.org/10.12686/alparray/z3_2015), RESIF-RLBP French broadband network

(http://doi.org/10.15778/RESIF.FR), Regional Seismic Network of North-Western Italy (https://doi.org/10.7914/SN/GU),

Mediterranean Very Broadband Seismographic Network (MedNet) (https://doi.org/10.13127/SD/FBBBTDTD6Q), and the

Italian National Seismic Network (http://doi.org/10.13127/SD/X0FXnH7QfY). For calculations, we use the python-based tool

ObsPy (Beyreuther et al., 2010; Krischer et al., 2015) and MATLAB (https://de.mathworks.com/). Figures were created using

Generic Mapping Tools version 6 (Wessel et al., 2019), MATLAB, and Inkscape (www.inkscape.org). We thank Thomas

Meier and his working group at Kiel University for their help with the multiple filter technique calculations and feedback on

the dispersion curves. Further, we thank the AlpArray Seismic Network Team, a complete list of members can be found here:

http://www.alparray.ethz.ch/en/seismic_network/backbone/data-policy-and-citation/. We thank two anonymous reviewers and

editor Mark Handy for their fruitful comments. Also, we thank Louisa Murray-Bergquist for checking the language of our

manuscript.

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
