# Peer review of "3D Crustal Structure of the Ligurian Sea Revealed by Surface Wave Tomography using Ocean Bottom Seismometer Data"

_Solid Earth, 2021_

## Referee Comment (RC1)

**Review of:**
**"3D crustal structure of the Ligurian Sea revealed by ambient noise tomography using ocean bottom seismometer data"**

Felix Noah Wolf[1], Dietrich Lange[1], Anke Dannowski[1], Martin Thorwart[2], Wayne Crawford[3], Lars
Wiesenberg[2], Ingo Grevemeyer[1], Heidrun Kopp[1,2], and the AlpArray Working Group[+]

Overall this is a high quality manuscript that provides information on additional pre-processing requirements for OBS data, and performs a group and shear velocity inversion using ambient noise data and teleseisms. The authors use a new OBS dataset (LOBSTER) that is located in relatively shallow water compared to previous studies (~1000 – 2000 m depth) and find the fundamental mode is not always the most prominent signal requiring a more involved processing method. The resulting maps are used to interpret the lithospheric structure beneath the Ligurian sea and Alpine Region with a particular focus on Moho depth and sediment thickness variations and use the velocity variations to determine no evidence for mantle serpentinization.

Although the paper is reasonably well written (particularly the methods section), there are areas where more information and clarity are needed, particularly for the resolution and interpretation sections. Further resolution tests are needed to determine whether the features discussed are resolvable both laterally and in depth. This is particularly important for the teleseismic dataset and periods >20s. Depth sensitivity and resolution are not discussed. The results and interpretation raise some interesting discussion points on the variations in the lithospheric structure with a focus on sediment thickness changes and Moho depth variations, however both sections are confusing to follow and the interpretation often needs more explanation. The main problem is the use of surface waves as a direct measure for Moho depth. Surface waves should only be a proxy for discontinuities due to their broad depth sensitivity. Combining these indirect measures with more direct measure for Moho depth such as receiver functions would aid the authors interpretation and be a more compelling argument, coupled with a rigorous depth resolution analysis.

Below I delineate the issues:

Scientific questions and comments

The title suggests only ambient noise and OBS data are used. I suggest changing this to be more generic due to the extensive role of the teleseisms and also acknowledge the land based station. The Alpine region is also interpreted so should be included with the Ligurian Sea.

Abstract
Line 14 – State what velocity models used

Introduction – Add a review of the current knowledge of crustal thickness and sediment thickness. This then gives context to the interpretation later.
Line 40 – Be more specific than narrow and steep.

Methods

Lines 76-78 - 22 OBS/BBOBS are plotted in Figure 1 which I assume are the 22 with complete datasets. Given 28 were recovered, please comment on why the other 6 were not used. Potentially plot all the locations on Figure 1 but identify which were not used in this study and which were not recovered to see the full extent of the experiment.

Line 104 – Given ambient noise and ambient noise tomography have been obtained for studies going back to 2007, the statement saying there is "no established routine for ambient noise analysis" seems contradictory. For  example the RHUM-RUM, PI-LAB and VoiLA experiments all produced ANT results. Even on land, multiple processing techniques are applied. Suggest removing and focus on the fact OBS data require tilt and compliance removal as a crucial processing step and should be performed in future.

Lines 134-135 – Please explain why only select land-ocean pairs were used rather than all possible combinations. Did using the full dataset bias the directionality of the ray paths? Changing Figure 2 to a hit count for the ray paths would show this visually.

Line 189 – Given there is no overlap between the ambient noise and teleseisms, how confident can you be that the dispersion curves are complementary? Particularly for the ambient noise OBS-OBS pairs, they don't look complimentary in Figure 5a (b and c are good). Studies have also suggested that a $\pi/4$ phase shift is required when combining ambient noise and teleseisms (e.g., Boschi et al., 2013; Yao et al., 2006; Tsai, 2009; Tsai, 2010). Do you require this here?

Please comment on the criteria used in order to keep a station-station cross correlation pair. Was there a minimum signal to noise ratio required, or a certain number of days in a stack required?

Line 207 – What is the spacing of the new refined grid.

Line 208 – Why remove 11, 13 and 14s period?

Line 219 – What size were the checkers and for which periods? Minimum and maximum period for both methods should be shown to give an idea of the resolution across all periods.

Line 226 – Specify exact depth range for PREM (>4km depth?) and the same for Dannowski et al. 2020 (Surface – 4 km depth?).

The authors suggest shallow water depth and seasonal variations for the OBS may have negatively impacted the quality of the CCFs. Did the authors correct for the seasonal component or was this this not possible given the short deployment? Did you look at in the noise quality and whether there is a clear variation between the quality of deeper OBS and those shallower?

Adding more detail for the checkerboard tests and further resolution tests at the end of the data resolution section is required. Moving some of the checkerboard details from the results into this section would also improve clarity. Add synthetic resolution tests to see if the basin structure and sediment thickness can be recovered. Depth resolution tests are required such as spike tests and checkerboard tests.

Results

The results section and interpretation are confusing. Currently the results are focussed on some error analysis and an overview and interpretation of the group velocities. The interpretations section then does the same for the shear velocities. I suggest changing this section to include the results from the 1-D dispersion curves, group velocities and finally shear velocities and move all interpretation to the next section.

In this section state what periods you are going to interpret for the group velocity maps (e.g. 5 – 40s) and also for the shear velocity maps. It would also be useful near the start of the results to give a broad overview of the range in velocity for the region and for the minimum and maximum period/depth.

The authors suggest the resolvable area has been chosen based on the checkerboard tests. However for 40s period the recovered checkerboard is much smaller than the area shown. Line 264 suggests one polygon is drawn for all periods, however this area seems only relevant for the shortest periods (5-8s) and is not appropriate for any of the teleseismic periods (>20s). I suggest the authors produces further resolution tests at a larger range of periods including the minimum and maximum periods used. Also state the size of the checkers (it looks like +- 1 degree) and show other tests with increased checkers for longer periods, as currently 40s is not well resolved. With the current tests I cannot tell whether there is resolution for teleseisms at the shorter periods (e.g. 20s). This should also move to the previous section.

Line 280 – Please comment on the crustal thickness of the area in the introduction and give the reader context on what depths the 20 and 40 s period are sensitive to.

Line 289 – Again give the reader information about the crustal thickness for the area. This is also a bold statement. How have you ruled out compositional variations, temperature and grain size. This is also interpretation so should probably go later in the manuscript.

Variations in the group and shear velocity are smaller than the checkerboard tests and alternate from slow to high velocity laterally. This may suggest the grid is too fine scaled and producing artefacts. Checkerboards at a finer scale would help determine if these are real features.

Interpretation

Lines 310-311 – This sentence currently reads as though the Group and shear velocities show different things. I don't think you mean this.

Line 312 – Does the average RMS vary with depth? Given much of the discussion is focussed on Moho depths if would be useful to see how RMS varies with depth with it likely to be better at shallow depths and worse at greater depths aligned with the Moho. Showing the resolution matrix would be useful to assess the resolution with depth.

Lines 314-317 – Move to results section.

Line 318 – Too strong to say it is definitely the geology at this point. I agree it is most likely variations in the geology and composition but you need to prove it. Suggest your hypothesis

then build up the evidence with where the sediments are located and the more mafic compositions. Can you rule out temperature effects, grain size and fluids for your study?

Line 346 – Please provide a reference and sentence summarising the work of geological studies to back up your assumption of sediments. Remove the word assumption.

Line 353 – Be clear whether these magmatic intrusions are recent or ancient. If they are recent and still contain melt then shear velocity will decrease, whereas if they are solidified then as you say, velocity will increase. I am also surprised that the velocities are not higher if they were solidified mafic intrusions. What is the rock type associated with the magmatic intrusions?

Line 355-368 – This paragraph is an over interpretation. Surface waves are sensitive to a broad range of depths particularly with increasing depth and are unlikely to characterise a distinct Moho. All of the shear velocity maps are shown over a depth range of 3km (e.g. 9-12 and 12-15 km). Without resolution tests for depth showing that an exact 1km layer is resolved, the authors cannot comment on observing the Moho. I suggest further resolution tests to aid this, combined with methods that are direct measures of discontinuities such as receiver function analysis. Cross sections for the study area are also required.

Line 359 – Where have you taken these velocity values from? Are they the range for the resolved area of mantle or for 1 point beneath the centre of the basin? More details required.

Line 413-424 – You state that the Alpine front should be visible but also say it is only 50km wide. Given checkerboards tests show shallow features are resolvable at 1 degree is it possible to resolve this feature?

Conclusions
The conclusions and abstract are clear and summarise the key points of the study very well.

Figures
Figure 1 – Overall a clear figure. I suggest colour coding or using different symbols for the various networks in order to distinguish them. Enlarge the inset map to a scale where the red square is more visible. It might be worth using a different colour box for the tectonic features to differentiate from countries, or enlarge the text for the countries. The same is true for the instruments. Also explain why only some instruments are highlighted. 22 OBS/BBOBS are plotted which I assume are the instruments with complete datasets.

Figure 2 – Potentially colour the ray paths according to hit count. This would also help determine if only using select OBS-Land combinations is appropriate. Label images as ambient noise and teleseisms respectively.

Figure 3 – It would be useful to see a land-land ambient noise pair. Perhaps if space is an issue, include in the supplementary material a good and bad version for each cross correlated pair for each combination of stations (e.g. land-land, OBS-OBS etc). The caption suggests one pair is not included which I think is b) but if c) was also discussed, this would be clearer.

Figure 4 – Looks ok

Figure 5 – Space needed in group velocity but otherwise looks good.

Figure 6 – Draw a line around areas with no ray coverage and low resolution to make clearer. In caption say purple triangles are stations. Enlarge the font size for the depth.

Figure 7 – Please include checkerboard tests for the shortest and longest periods used for both ambient noise (5 -15s) and the teleseisms (20 - 90s or 20 – 40s if these are the only interpreted images) to give a clear indication of the extent of resolvable features. If space is an issue add to the supplementary material. What size are the anomalies? Please state. Enlarge the font size for the depth. Also include checkerboards for different scales.

Figure 8 – Enlarge the font size for depth.

Data Availability – While the raw data is accounted for, the final model is not included or any outputs for the processed cross correlation functions. I recommend uploading at least the final model with errors as a supplementary material in line with the journals policies.

Technical Corrections
Line 25 – Change "We find no hint on mantle serpentinization …" to "We do not observe mantle serpentinization…"
Line 33 – Remove the use of "on."
Line 41 – 42 – This sentence is not clear. Perhaps commas would help?
Line 94 – Add e.g. before references. There are many papers that have developed the ANT technique.
Line 203 – Add reference to figure 7 (checkerboard tests).
Line 286 – Remove "an"
Line 295 – Section 4.2 I think should be 5.2
Line 286 – Remove "an" in "with  increasing period"
Line 328 – Change "fits" to "is comparable"
Line 348 – Reference required for the sediment thicknesses

---

## Author Comment (AC1)

**Answers to reviewer comments #1**

https://doi.org/10.5194/se-2021-55-RC1

**Title: 3D crustal structure of the Ligurian Sea revealed by ambient noise tomography using ocean bottom seismometer data**

Dear Reviewer,

We thank you for the fair, detailed and constructive review of our manuscript se-2021-55. Please find below our line-by-line-revision where we carefully respond to all your comments and suggestions and explain how we modified the manuscript.

The comments from the reviews are indicated in **black** and our replies in **red** colour.

Scientific questions and comments

The title suggests only ambient noise and OBS data are used. I suggest changing this to be more generic due to the extensive role of the teleseisms and also acknowledge the land-based station. The Alpine region is also interpreted so should be included with the Ligurian Sea.

We changed the title accordingly by adding the words "surface wave tomography" which includes both, the ambient noise and the teleseismic cross-correlations. We stick to "Ligurian Sea" because we see this as the focus of the manuscript. Also, we did not change "ocean bottom seismometer", because we want to emphasize the offshore part of the dataset. The land stations are clearly marked and acknowledged in the manuscript, but the vast majority of new conclusions of this paper is resulting from the use of OBS (ocean bottom seismometer) data.

Abstract

Line 14 – State what velocity models used

We updated this sentence to prevent misunderstandings on the work done. We did not invert velocity models, but performed surface wave tomographies.

Old: "We invert velocity models using an amphibious network [...]"

New: "*We perform ambient noise tomography, also taking into account teleseismic events, using an amphibious network [...]*"

Introduction – Add a review of the current knowledge of crustal thickness and sediment thickness. This then gives context to the interpretation later.

We added a short paragraph on the sediment thickness and a longer one about the knowledge of the crustal thickness.

New: "*The marine bedrock is covered by a sediment layer (e.g. Schettino and Turco, 2006) of varying thickness. It is less than 3 km thick near the Tuscany coast, increases towards the southwest, and reaches a thickness of up to 8 km offshore Marseille.*"

Line 40 – Be more specific than narrow and steep.

We specified: "*The continental margin is narrow (10-20 km) and steep at the Ligurian coast (Finetti et al., 2005) and broader (20-50 km) on the Corsican side (e.g. Rollet et al., 2002)."*

Methods

Lines 76-78 - 22 OBS/BBOBS are plotted in Figure 1 which I assume are the 22 with complete datasets. Given 28 were recovered, please comment on why the other 6 were not used. Potentially plot all the locations on Figure 1 but identify which were not used in this study and which were not recovered to see the full extent of the experiment.

We use all stations for which all three seismometer components and the hydrophone component worked correctly during the complete deployment. This is true for the 22 OBS shown. Showing stations that were not used (no data or not enough data was recorded) would, in our opinion, not increase the clarity of the figure. Therefore, the corresponding numbers in Section "Data" were updated.

Line 104 – Given ambient noise and ambient noise tomography have been obtained for studies going back to 2007, the statement saying there is "no established routine for ambient noise analysis" seems contradictory. For example, the RHUM-RUM, PI-LAB and VoiLA experiments all produced ANT results. Even on land, multiple processing techniques are applied. Suggest removing and focus on the fact OBS data require tilt and compliance removal as a crucial processing step and should be performed in future.

We changed "there is no established routine" to "it is not yet used regularly".

Lines 134-135 – Please explain why only select land-ocean pairs were used rather than all possible combinations. Did using the full dataset bias the directionality of the ray paths?

Of course, we use all land-ocean pairs. To clarify this, we deleted the sentence and rephrased the paragraph to clarify that we correlated all OBS-OBS and all OBS-land stairs. The selection of additional land-land pairs was made to increase the ray coverage in the Ligurian basin.

Old version: "We use the tilt- and compliance-corrected daily files to estimate cross-correlation functions (CCF) for every vertical component OBS-OBS station pair (Bensen et al., 2007). These CCFs are complemented by CCFs for selected OBS-land station pairs and land-land pairs to obtain cross-correlations in other ray directions than those of the OBS-land pairs (Fig. 2). Additionally, we calculate CCFS for all land-land pairs for the land stations A317A, ARBF, and DIX (see Fig. 1) and CCFs for all combinations of 20 selected land stations (namely AJAC, BLAF, BOB, BSTF, CALF, CARD, EILF, ENR, GBOS, IMI, ISO, MSSA, PCP, PLMA, ROTM, SAOF SMPL, TRBF, TURF, and VLC) to increase the resolution onshore. "

New: "*We use the tilt- and compliance-corrected daily files to estimate cross-correlation functions (CCF) for every vertical component OBS-OBS and OBS-land station pair (Bensen et al., 2007). Additionally, we calculate CCFS for all land-land pairs for the land stations A317A, ARBF, and DIX (see Fig. 1) and CCFs for all combinations of 20 selected land stations (namely AJAC, BLAF, BOB, BSTF, CALF, CARD, EILF, ENR, GBOS, IMI, ISO, MSSA, PCP, PLMA, ROTM, SAOF SMPL, TRBF, TURF, and VLC) to increase the ray coverage onshore (Fig. 2).*"

Changing Figure 2 to a hit count for the ray paths would show this visually.

We updated Figure 2 accordingly.

Line 189 – Given there is no overlap between the ambient noise and teleseisms, how confident can you be that the dispersion curves are complementary? Particularly for the ambient noise OBS-OBS pairs, they don't look complimentary in Figure 5a (b and c are good). Studies have also suggested that a $\pi/4$ phase shift is required when combining ambient noise and teleseisms (e.g., Boschi et al., 2013; Yao et al., 2006; Tsai, 2009; Tsai, 2010). Do you require this here?

The OBS-OBS dispersions not looking that complimentary might be caused by the comparably low number of pairs; a great portion of the dispersion curves do look complementary, but 'outliers' are more visible due to the low number of curves.

The mentioned phase-shift applies to investigations of phase velocity (e.g. the mentioned Boschi et al, 2013). Since we derive group-velocities from cross-correlation functions in the time-domain, we do not require a phase shift for teleseismic cross-correlations.

Please comment on the criteria used in order to keep a station-station cross correlation pair. Was there a minimum signal to noise ratio required, or a certain number of days in a stack required?

We manually picked the dispersion curves. Station pairs, where no correlation maximum was detectable in the MFT plot, were excluded. To clarify, we added a sentence in line 194: "*Station pairs showing no detectable maximum were excluded.*"

Line 207 – What is the spacing of the new refined grid. added

Line 208 – Why remove 11, 13 and 14s period?

We did not "remove" 11 s, 13 s, and 14 s period, but decided to follow common practice and apply less sampling at increasing periods, as we also do for periods > 15 s. We did not modify the manuscript.

Line 219 – What size were the checkers and for which periods? Minimum and maximum period for both methods should be shown to give an idea of the resolution across all periods.

We added the periods and sizes to the caption of Figure 7 (checkerboard tests). We also added more checkerboard tests to the supplementary material (Fig. S4-S7).

Line 226 – Specify exact depth range for PREM (>4km depth?) and the same for Dannowski et al. 2020 (Surface – 4 km depth?).

We added the missing depth range.

Old: "[...] we set up a starting model with fixed layers (Table S4) that is based on PREM (Dziewonski and Anderson, 1981) at depth and on $v_P$-velocities from Dannowski et al. (2020) for the shallow layers."

New: "*[...] we set up a starting model with fixed layers (Table S4) based on the $v_P$-velocities from Dannowski et al. (2020) for up to 16 km depth and on PREM (Dziewonski and Anderson, 1981) below.*"

The authors suggest shallow water depth and seasonal variations for the OBS may have negatively impacted the quality of the CCFs. Did the authors correct for the seasonal component or was this not possible given the short deployment? Did you look at the noise quality and whether there is a clear variation between the quality of deeper OBS and those shallower?

We looked into the data and observed more days with high noise levels during the later autumn and winter months. The quantitative evaluation of the influence of the seasonal signal on the CCFs would be out of the scope of this paper. Therefore, we changed the sentence "and the highly variable weather conditions of our research area" to "*and probably also the highly variable weather conditions of our research area*".

A seasonal effect was not removed, due to the deployment time of only 8 months. The seasonal component can be removed more easily for data sets that are longer than a year.

Adding more detail for the checkerboard tests and further resolution tests at the end of the data resolution section is required. Moving some of the checkerboard details from the results into this section would also improve clarity. Add synthetic resolution tests to see if the basin structure and sediment thickness can be recovered. Depth resolution tests are required such as spike tests and checkerboard tests.

We added details on the checkerboard tests here. We increased the number of tests and varied the grid size. Also, we calculated a restoration test for $v_G$- and $v_S$-maps to increase the understanding of the resolvable areas.

Results

The results section and interpretation are confusing. Currently the results are focussed on some error analysis and an overview and interpretation of the group velocities. The interpretations section then does the same for the shear velocities. I suggest changing this section to include the results from the 1-D dispersion curves, group velocities and finally shear velocities and move all interpretation to the next section.

In this section state what periods you are going to interpret for the group velocity maps (e.g. 5– 40s) and also for the shear velocity maps. It would also be useful near the start of the results to give a broad overview of the range in velocity for the region and for the minimum and maximum period/depth.

Thank you for this feedback. We carefully reorganized the results and interpretation sections. Please see the track changes version for the in-depth revisions of both sections.

The authors suggest the resolvable area has been chosen based on the checkerboard tests. However, for 40 s period the recovered checkerboard is much smaller than the area shown. Line 264 suggests one polygon is drawn for all periods, however this area seems only relevant for the shortest periods (5-8s) and is not appropriate for any of the teleseismic periods (>20s). I suggest the authors produces further resolution tests at a larger range of periods including the minimum and maximum periods used. Also state the size of the checkers (it looks like +- 1 degree) and show other tests with increased checkers for longer periods, as currently 40 s is not well resolved. With the current tests I cannot tell whether there is resolution for teleseisms at the shorter periods (e.g. 20s). This should also move to the previous section.

We increased the number of resolution tests and information in the manuscript. We updated the checkerboard tests (CB) in section 4.1 (now 5 s, 15 s, 20 s, 90 s). Checkerboards for all other periods are calculated as well and are shown in the supplement. We added CBs with a finer grid size (0.4°x0.4° in addition to 0.8°x0.8°) to give more information about our resolution for small scale anomalies. Additionally, we now include a restoration test based on synthetic data (Figures in the Supplement). We also added a section on resolution tests to the previous section on data quality (3.4).

Line 280 – Please comment on the crustal thickness of the area in the introduction and give the reader context on what depths the 20 and 40 s period are sensitive to.

We added a paragraph on the crustal thickness to the introduction. The discussion on sensitivity was added and moved to the discussion.

New (Section 5.2): "*This hints at less heterogeneities in the mantle, but might also be caused by the decreasing sensitivity of group velocities with increasing period (e.g. Adimah & Padhy, 2020). For a period of 5 s, the group velocity is sensitive to a narrow depth range that peaks at ~5 km. For 20 s, the overall sensitivity is lower and has a much broader range of approximately 10-25 km depth.*"

Line 289 – Again give the reader information about the crustal thickness for the area. This is also a bold statement. How have you ruled out compositional variations, temperature and grain size? This is also interpretation so should probably go later in the manuscript.

This sentence was deleted while updating the manuscript.

Variations in the group and shear velocity are smaller than the checkerboard tests and alternate from slow to high velocity laterally. This may suggest the grid is too fine scaled and producing artefacts. Checkerboards at a finer scale would help determine if these are real features.

Finer checkerboards were added. We tested different grid sizes and used the one with the lowest overall RMS. Artefacts do appear in the north of our research area (fast area "fingers") and between Corsica and the Italian coast. Both are caused by the ray coverage. The "fingers" offshore Nice are discussed in Sect. 5.2.

Interpretation

Lines 310-311 – This sentence currently reads as though the Group and shear velocities show different things. I don't think you mean this.

Indeed, we do not mean that. We rephrased this sentence to increase clarity.

Old: "*We calculated 1D depth inversions for $v_S$ based on the group velocity maps described above. In the following, we discuss four distinct regions that show different characteristics in the shear velocity maps.*"

New: "*In the following, we discuss three regions that show differing characteristics in the velocity maps: [...]*"

Line 312 – Does the average RMS vary with depth? Given much of the discussion is focussed on Moho depths it would be useful to see how RMS varies with depth with it likely to be better at shallow depths and worse at greater depths aligned with the Moho. Showing the resolution matrix would be useful to assess the resolution with depth.

The inversion outputs one RMS value foreach inverted 1D profile. We now show the RMS values of the 1D inversion in mapview (Figure 8, panel i).

Lines 314-317 – Move to results section. Done.

Line 318 – Too strong to say it is definitely the geology at this point. I agree it is most likely variations in the geology and composition but you need to prove it. Suggest your hypothesis then build up the evidence with where the sediments are located and the more mafic compositions. Can you rule out temperature effects, grain size and fluids for your study?

We are well aware that changes in temperature, grain size and fluids affect the velocity. Nevertheless, the variations in geology include changes of these parameters and we did not try to estimate the effect of each one individually. In order to clarify:

New: "*In up to 9 km depth (Fig. 8a-c), we observe laterally varying shear-wave velocities on land that we assume to be caused by variations in the geology. At the Rhône delta (Fig. 1), where the sediment cover is up to 12 km thick (Pichon et al., 2010), we observe $v_S \cong 2.7$ km s$^{-1}$ in the 4-6 km layer (Fig. 8b) and $v_S \cong 3$ km s$^{-1}$ in 6-9 km depth (Fig. 8c). Similarly, the Po plain has an average sedimentary cover of 7-8 km (Molinari et al., 2015a) with a shear-wave velocity increasing from $v_S \cong 2.5$ km s$^{-1}$ to $v_S \cong 3.1$ km s$^{-1}$ at 4-9 km depth. In contrast to the sediment basins, we observe higher $v_S \cong 3-3.5$ km s$^{-1}$ (4-9 km depth) beneath the Alpine belt, composed of crystalline and metamorphic rocks (e.g. Molinari et al., 2015b). This S-wave variation is, most probably, caused by the different geology of the Alpine belt and the sedimentary basins.*"

Line 346 – Please provide a reference and sentence summarising the work of geological studies to back up your assumption of sediments. Remove the word assumption.

We rephrased the sentence:

Old: "*We assume that sediments mainly dominate the S-wave velocity for the shallow layers.*"

New: "*At shallow depths (Fig. 8a-b), the S-wave velocity is mainly dominated by sediments.*"
A reference and summary of geological studies is already given in the manuscript, following the rephrased sentence.

Line 353 – Be clear whether these magmatic intrusions are recent or ancient. If they are recent and still contain melt then shear velocity will decrease, whereas if they are solidified then as you say, velocity will increase. I am also surprised that the velocities are not higher if they were solidified mafic intrusions. What is the rock type associated with the magmatic intrusions?
We added a short paragraph on the volcanism in the Ligurian Basin to the introduction. The youngest volcanism north of Corsica and in the Gulf of Genova dates back to 12-11 Ma (Rollet et al., 2002).

Line 355-368 – This paragraph is an over interpretation. Surface waves are sensitive to a broad range of depths particularly with increasing depth and are unlikely to characterise a distinct Moho. All of the shear velocity maps are shown over a depth range of 3km (e.g. 9-12 and 12-15 km). Without resolution tests for depth showing that an exact 1km layer is resolved, the authors cannot comment on observing the Moho. I suggest further resolution tests to aid this, combined with methods that are direct measures of discontinuities such as receiver function analysis. Cross sections for the study area are also required.
We agree that we can not resolve a discontinuity with the method. We rephrased the section to make it clear that we rely on Moho depth estimates based on seismic data. Given the depth along seismic lines, we then use our data to make assumptions on the 3D structure.

Line 359 – Where have you taken these velocity values from? Are they the range for the resolved area of mantle or for 1 point beneath the centre of the basin? More details required.
For clarification, we added: "*[…] at the southwestern end of their profile*".

Line 413-424 – You state that the Alpine front should be visible but also say it is only 50 km wide. Given checkerboards tests show shallow features are resolvable at 1 degree is it possible to resolve this feature?
We rewrote this section and also added checkerboard tests with a finer grid size.

Conclusions
The conclusions and abstract are clear and summarise the key points of the study very well.
Thank you very much for your constructive and positive comments.

Figures
Figure 1 – Overall a clear figure. I suggest colour coding or using different symbols for the various networks in order to distinguish them. Enlarge the inset map to a scale where the red square is more visible. It might be worth using a different colour box for the tectonic features to differentiate from countries, or enlarge the text for the countries. The same is true for the instruments. Also explain why only some instruments are highlighted. 22 OBS/BBOBS are plotted which I assume are the instruments with complete datasets.
We added the OBS stations that were not used to the map, color coded in red. Also, we enlarged the text for the countries and stations and chose a smaller area for the inlay map, so the red square is more visible. All OBS are named, but on land we only named the three stations, for which we calculated all land- land cross-correlations (A317A, ARBF, DIX), as well as the station that is part of the exemplary OBS-land pair in Fig. 5 (BHB).

We decided not to use different symbols for the different land station networks. In our opinion, this would not increase the clarity of the figure. The network code, name, and location of all stations are listed in Table S1.

Figure 2 – Potentially colour the ray paths according to hit count. This would also help determine if only using select OBS-Land combinations is appropriate. Label images as ambient noise and teleseisms respectively.
We followed this suggestion and updated Figure 2, now showing hit counts. The caption already states that 8 s and 20 s show the ray coverage of ambient noise cross-correlation and teleseismic cross-correlation, respectively.

Figure 3 – It would be useful to see a land-land ambient noise pair. Perhaps if space is an issue, include in the supplementary material a good and bad version for each cross correlated pair for each combination of stations (e.g. land-land, OBS-OBS etc). The caption suggests one pair is not included which I think is b) but if c) was also discussed, this would be clearer.
We updated the caption to make clear that (b) was excluded. Also, we added a land-land station pair. We did not include a bad station pair because we think it does not improve the clarity and focus of the manuscript.

Figure 4 – Looks ok

Figure 5 – Space needed in group velocity but otherwise looks good. Corrected.

Figure 6 – Draw a line around areas with no ray coverage and low resolution to make clearer. In caption say purple triangles are stations. Enlarge the font size for the depth.
Updated accordingly.

Figure 7 – Please include checkerboard tests for the shortest and longest periods used for both ambient noise (5 -15s) and the teleseisms (20 - 90s or 20 – 40s if these are the only interpreted images) to give a clear indication of the extent of resolvable features. If space is an issue, add to the supplementary material. What size are the anomalies? Please state. Enlarge the font size for the depth. Also include checkerboards for different scales.
We changed Figure 7. Now, we show checkerboards for 5 s, 15 s, 20 s and 90 s. Checkerboards for all other periods are added to the supplementary material (Fig. S4-S7). We now show one checkerboard with 0.2°x0.2° grid size as well as one with 0.8°x0.8° grid size for each period. Fonts are enlarged.

Figure 8 – Enlarge the font size for depth.
We updated Fig. S3 accordingly. We also added a figure showing the root mean square (RMS) value for every grid cell for the shear-wave-velocity inversion.

Data Availability – While the raw data is accounted for, the final model is not included or any outputs for the processed cross correlation functions. I recommend uploading at least the final model with errors as a supplementary material in line with the journals policies.
We now provide a *.zip-folder containing the in- and output of the group velocity tomography, the code we use to create 1D shear-wave velocity inversion input files from that, and the resulting output as *.xyz files (including plot-script).

Technical Corrections

Line 25 – Change "We find no hint on mantle serpentinization …" to "We do not observe mantle serpentinization…" We changed this.

Line 33 – Remove the use of "on." removed

Line 41 – 42 – This sentence is not clear. Perhaps commas would help? We added commas to structure the sentence.

Line 94 – Add e.g. before references. There are many papers that have developed the ANT technique. We added "e.g.".

Line 203 – Add reference to figure 7 (checkerboard tests). added

Line 286 – Remove "an" removed

Line 295 – Section 4.2 I think should be 5.2 Well spotted, thanks! We changed that.

Line 286 – Remove "an" in "with an increasing period" removed

Line 328 – Change "fits" to "is comparable" changed

Line 348 – Reference required for the sediment thicknesses reference added

---

## Author Comment (AC2)

**Answers to reviewer comments #2**

https://doi.org/10.5194/se-2021-55-RC2

**Title: 3D crustal structure of the Ligurian Sea revealed by ambient noise tomography using ocean bottom seismometer data**

Dear Reviewer,

We thank you for the fair and constructive review of our manuscript se-2021-55. Please find below our line-by-line-revision where we carefully respond to all your comments and suggestions and explain how we modified the manuscript.
The comments from the reviews are indicated in **black** and our replies in red colour.

**ABSTRACT**
L26: "no hint OF mantle…"
Changed accordingly.
**INTRODUCTION**
L37: The term "collision" is improper for describing the emplacement of the Sardinia-Corsica block in its current position. Rather, this block is a crustal remnant left behind during the opening of the Tyrrhenian Sea.
We rephrased the sentence according to the editor's suggestion:
"*the Corsica-Sardinia block was stranded between the Apennines and the European margin in southern France*".

L57: In reality, the paper mainly focuses on the results and does not investigates the geodynamic implication of these (will it be part of a later work?). Thus, I'd rephrase the "To better understand the evolution of the Ligurian Basin and the processes driving its formation". We rephrased this to: "*To better understand the present-day crustal velocity structure and its implications on the evolution of the Ligurian Basin, [...].*"

FIGURE1: Remove the decimal digits in thick labels. In the map in the bottom-left corner, just show the European continent.
Thank you for these suggestions. We updated Figure 1 accordingly. In the inlay-map, we choose a smaller area centred at the OBS array.
**DATA**
L76: Can you explain why some of the stations/data were not recovered? What were the issues?
We decided not to mention the stations that we did not use. Instead, we only focus on the 22 stations that did record all components for the complete deployment, in order to increase the focus of the manuscript. The other stations were not used, because they did not record (enough) data.

**METHOD**
L113: "horizontal signal…" —> horizontal movement We corrected this.

Majority of readers are used to land data. It'd be interesting to show an example of OBS data before and after tilt/compliance correction. Would you please provide that?

A very nice example of OBS data before and after removing tilt and compliance is shown in Figure 9 of Crawford and Webb, 2000.

FIGURE 2: please use a color for the stations that gives more contrast with the background (red/white). Tip: you can use shaded black lines (transparency set to 0.3-0.5) for the line connecting the stations. This will immediately gives an idea of ray density…
We followed the suggestion of reviewer #1 and created hit count maps to show the ray coverage.

FIGURE 4: there is a remnant label "180"…
We could not find the remnant label. Therefore, Figure 4 is unchanged.

L188: In the frequency range that is common to both ambient noise and teleseismic events, what did you do to calculate the group velocity? Average? Weighted-average? According to several papers, when using teleseismic events one systematically overestimate group/phase velocity (several explanation have been hypothesized as a reason for this, fist of all: off-path arrivals). It might be worth having a look to Magrini et al. 2019 (10.1093/gji/ggz560) where they deal with this issue.
Thank you for this comment. To clarify, we did not use overlapping frequency ranges. We decided to evaluate the ambient noise group velocities for periods of 5 – 15 s, and the teleseismic group velocities for periods of 20 – 90 s.
Magrini et al. (2019) is a very interesting study and they do improve the phase velocity results by a significant, though small amount. The change in velocity Magrini et al. observe is in the range of 0.04 km/s. Since we are using group velocities to estimate shear wave velocities, and interpret differences of > 0.1 km/s, we prefer not to apply their method.

FIGURE 5: y-axis label: put a space between "group" and "velocity". Use some transparency for the curve to make them all visible…What is the error on the group velocity estimation? How is it estimated? Can you shrink the x-axis? The curve is not really appreciable and seems really flat.
We corrected the typo and shrank the x-axis, to make it easier to see that the curves are not flat. The error is not estimated based on the dispersion curves but on the picked lagtimes.

L223: Can you provide some more details on the inversion code? Is it stochastic or linearized? Are data uncertainties accounted for and, if yes, how?
Added "iterative, weighted inversion [code]". The uncertainties are accounted for.

**RESULTS**
FIGURE 6: show colorbar only once and make it bigger We updated Figure 6 accordingly.

FIGURE 7: again, show colorbar only once and make it bigger
We updated Figure 7 accordingly. We also included several more checkerboard setups to allow more conclusions on the resolution. More checkerboard tests are shown in the supplementary material.

FIGURE 8: "depth inversion"?. It's rather an inversion from group-velocities to shear-wave velocities. There's no need to repeat the colorbar 8 times if it is the same.
That is a good point. We invert for shear-wave velocities and keep the depth layers fixed. Therefore, we followed your suggestion and changed the naming. Also, we updated Figure 8.

L313: Why is the RMS here? Please move it to the method or result part. moved to results

**DISCUSSION**

Here I suggest to comment further on the implication of the results obtained. For example: if there's no serpentined mantle in the basin centre, what causes such high velocities directly underneath the sediments? Can these be caused by thick oceanic crust? What does that mean in the context of the evolution of the Liguro-Provençal basin?

We restructured and re-wrote significant parts of the discussion section. Also, more details are given to the questions raised here. For example, we added a paragraph on the implications on the present day basin structure and its implications for its evolution at the end of Section 5.2: "*Dannowski et al. (2020) suggest that continental crust was (extremely) thinned along their profile, but no spreading occurred. This is in-line with our results. A possible spreading centre has to be located to the southwest. At the Gulf of Lion margin, at the southwestern edge of our research area, Gailler et al. (2009) interpreted their results as oceanic crust, also observing a transition zone made up of "lower crustal material or mixture of serpentinized upper mantle material with lower crustal material" (Gailler et al., 2009). Later, Jolivet et al. (2015) explained the shallow high velocities by exhumed lower crustal material. They also suggest partially serpentinised mantle. Therefore, a possible spreading center might have been located southwest of our research area, possibly as close as the Gulf of Lion margin.*" Another paragraph was added to Section 5.3: "*The apparent thickening of the continental crust towards the northeast is likely related to the position of the rotational pole of the opening of the Ligurian Sea during the Oligocene-Miocene. According to Speranza et al. (2002) and Gattacceca et al. (2007), the rotational pole was located in the northeastern Ligurian Sea at 43.5°N, 9.5°E. Therefore, the southwestern basin was more extensively opened, and the continental crust was thinned further than in the northeast.*"

---

## Author Comment (AC3)

**Answers to editor comments**

https://doi.org/10.5194/se-2021-55-EC1

**Title: 3D crustal structure of the Ligurian Sea revealed by ambient noise tomography using ocean bottom seismometer data**

Dear Topical Editor Mark Handy,

We thank you for the fair and constructive editing and review of our manuscript se-2021-55. We carefully revised the manuscript on all levels, restructuring the sections results and discussion, improving the clarity of the manuscript and also improving the language. Also, we had a native speaker check the manuscript. Please find below the responses to all points raised. The editor comments are indicated in **black** and our replies are in red colour.

Abstract
The abstract should have a better balance of methodology (compilation of methodological steps used to obtain the images) and interpretation (only the 2 final sentences contain interpretation). The final sentence should be rephrased, because it is unclear how separation of the SW and NE Ligurian Basin is related to the promoted prolongation (do you mean southward continuation?) of the Alpine front.
We reworked this part of the abstract and added more detail to the results.
Old: "The shear-wave velocity results show a deepening of the Moho from 12 km at the southwestern basin centre to 20-25 km at the Ligurian coast in the northeast and over 30 km at the Provençal coast. We find no hint on mantle serpentinisation and no evidence for an Alpine slab, at least down to depths of 25 km. However, we see a separation of the southwestern and northeastern Ligurian Basin that coincides with the promoted prolongation of the Alpine front."
New: "*The group velocity and shear-wave velocity results compare well to existing large-scale studies that partly include the study area. Onshore France, we observe a high-velocity area beneath the Argentera Massif, roughly 10 km below sea level. Our results, in addition to existing seismic profiles, expand the knowledge on seismic velocities in the Ligurian Basin, adding spatial information. In accordance with existing seismic studies, our shear-wave velocity maps indicate a deepening of the Moho from 12 km at the southwestern basin centre to 20-25 km at the Ligurian coast in the northeast and over 30 km at the Provençal coast. We see a separation of the southwestern and northeastern Ligurian Basin. We do not observe high crustal $v_P/v_S$ ratios as proxy for mantle serpentinisation in the southwestern Ligurian Bain.*"

L17: "Attentively"...what do you mean? You may mean "carefully".
Yes, we mean "carefully" and changed this accordingly.

Introduction
L34: Add more references, e.g., Dewey et al.1989, Sérrane 1999, Speranza et al. 2002, Schettino and Turco 2006, Le Breton et al. 2017 and many references therein.
We added more references and the prefix "e.g.", because many authors stated this before. Furthermore, we added a sentence referring to Le Breton et al. (2017).

L36: No way did it ever collide! It was stranded as a rift block (horst) between the Apennines and the European margin in S. France.

We agree and rephrased the sentence accordingly.

L38: That is a huge range; please add a sentence saying why.
We added a sentence that says that the Ligurian Sea broadens from the northeast to the southwest.
Old: "Today, the Ligurian Sea is 150-225 km wide, while the basin itself has a width of 70-170 km (Dannowski et al., 2020)."
New: "*Today, the Ligurian Sea is 150-225 km wide, while the basin itself has a width of 70-170 km (Dannowski et al., 2020), broadening from the northeast to the southwest. The continental margin is narrow (10-20 km) and steep at the Ligurian coast (Finetti et al., 2005) and broader (20-50 km) on the Corsican side (e.g. Rollet et al., 2002).*"

L39: Reference? How do you know?
We added references to Rollet et al., 2002, Finetti et al., 2005, and Dannowski et al., 2020 (see above).

L43: See also Schettino & Turco 2006 and papers since
Added the Schettino & Turco, 2006, reference and also added "e.g." to make clear that these references to this generic statement are not complete.

L43: Neighbor is a noun, not a verb! Better to say "This area is located next to..."
We rephrased this sentence as suggested.

L45: Better "...seafloor spreading did not occur..." rephrased accordingly.

L45: "...that beneath the SW part of the basin..." rephrased accordingly.

L49: Where in Italy? The Alpine front on the European continent is in southern France. In Italy, the orogenic front is of the Apennines, not the Alps.
We totally agree and changed this to "France".

L52: also Kaestle et al. 2020
We cite different ambient noise studies. Kästle et al. (2020) is a review paper, comparing existing studies. Therefore, we prefer to cite the original work, which is Kästle et al. (2018).

Results
Figure 6: Refer to discussion of this interpretation in the text.
We removed the dashed line in Figure 6. The proposed offshore prolongation of the Alpine front is discussed in Sect. 5.4.

Figure 8: As in Fig. 6, please mention a discussion of this interpretation in the text. The reader will want to know if you have just connected the onshore trace of the Alpine front, or if you have independent seismological criteria for drawing the line.
The dashed line only represents the proposed offshore front, connecting the onshore trace of the Alpine front. We added the Rollet et al. (2002) citation to make this clearer.

Discussion
L317: Reference? e.g., Bigi et al. 1989
We moved the citation to Molinari et al., (2015a) to clarify: "*Similarly, the Po plain has an average sedimentary cover of 7-8 km (Molinari et al., 2015a) [...]*"

L320: To save this entire section (lines 320-340), make a figure that allows the reader to visibly compare the images from the different works. You could save a lot of text that way. As it stands, one must read your long text and imagine the images of the previous work or have them printed out and laid side-by-side.

This is a good idea, however this comparison of the onshore regions is not the main focus of our study. We focus on the Ligurian Sea, where the other studies have only little resolution. We compare our onshore results to document the robustness of our tomography and inversion images. To make it easier to follow the comparison, we added figure numbers of the compared works in the text (e.g.: "*The large-scale structures also compare nicely to the results of Kästle et al. (2018, their Fig. 9) and Lu et al. (2018, their Fig. 7).*").

L352: The real reason might be the exhumation of dense upper mantle rock at or near the basin axis.
We rephrased these sentences.
Old: "We deduce that for fast areas along the basin axis, due to the thinning of continental crust, the velocity gradient is stronger than away from the basin axis. This would lead to a higher S-wave velocity near the basin axis."
New: "*We deduce that for fast areas along the basin axis, the velocity gradient is stronger than away from the basin axis. This is probably caused by the thinning of continental crust (Dannowski et al., 2020) and possible exhumation of denser lower crust and upper mantle rock (Gailler et al., 2009; Jolivet et al., 2015) observed further southwest. Both scenarios would lead to a higher S-wave velocity near the basin axis.*"

L375: "poor" instead of "sparce"
We changed the word accordingly..

L393: and that the mantle is closer to the surface.
This is true for the southwestern basin. For shallow layers in the northeastern basin, we link the increased $v_S$ (compared to the southwest) to thinner sediments and therefore to shallower bedrock. No changes were made in the manuscript.

L407: The exact location of the rotation pole of Corsica is a matter of debate and I would quote more than just one recent reference, unless you state good reasons to quote only Gattacceca et al. 2007). For example, see Seton et al. 2012, https://doi.org/10.1016/j.earscirev.2012.03.002
Seton et al. (2012) state that they base their reconstruction on Speranza et al. (2002). Therefore, we added the citation of Speranza et al. (2002). Speranza et al. (2002) use the same present-day coordinates as Gattacceca et al. (2007).

L414: If there has been even limited spreading in the Liguria Sea, then I would not expect the Alpine front to be continuous across this sea.
As Dannowski et al. (2020) state that a spreading center would be located further to the southwest of their profile, we assume no spreading this far northeast. Therefore, an offshore front could be detectable in the crust. No changes were made to the manuscript.

L415: Why beneath the crust? If the crust is merely attenuated, then the Alpine front (an Oligo-Miocene structure) would also be attenuated and may in fact be exposed throughout the crust and reach the seafloor.
We changed the whole section and took this into account. See comment on L421.

L416: subperpendicular.... This sentence was deleted while rewriting section 5.4 ("Alpine front").

L417: I would only expect an anomaly if the front juxtaposes very different types of rock. If not, then the front will not be imaged.
This sentence was also deleted.

L418: at high angles.... This sentence was also deleted.

L420: Please state where! This sentence was also deleted.

L421: There is a flaw in the argumentation here, because you implicitly equate the Alpine Front with a slab. The two are not the same. The front is merely the surface expression of the furthest (most external) and may have little if anything to do with a slab. The slab is the expression of subduction, but the expression of subduction at the surface is a suture (the occurrence of ophiolites in the mountain belt). This section needs fixing.
Thank you for pointing this out. We investigate the proposed offshore prolongation of the Alpine front, as proposed by Rollet et al. (2002). It might be detectable in the crust, existing seismic data and our results do not show evidence of it, though. We rewrote the whole section 5.4 accordingly:
*"Rollet et al. (2002) raised the question of an offshore prolongation of the Alpine front that can be observed onshore France and onshore Corsica. Rollet et al. (2002) suggested the Alpine front to separate the southwestern and northeastern parts of the Ligurian Basin. This proposed front is roughly located at the separation of the northeastern and southwestern crustal domains that we observe in our data (illustrated by the dashed line in Fig. 8e). However, the location and even existence of such a prolongation of the Alpine front beneath the Ligurian sea is not yet resolved.*
*As mentioned, the seismic records indicate no spreading this far northeast in the basin. Therefore, the proposed offshore Alpine front could be detectable in the crust. Dannowski et al. (2020) observe a gradual thickening of the continental crust towards the northeastern Ligurian Basin. They do not need the sharp step that Makris et al. (1999) introduced between Corsica and the Liguro-Provencal coast to explain the free-air anomaly derived by Sandwell et al. (2014). Our spatial shear-wave velocity data also supports that interpretation. We do not observe a sharp lateral boundary but also observe a gradual change of the velocity layers that fits the model of Dannowski et al. (2020). With the given resolution, an offshore prolongation of the Alpine front is not detectable."*

Author contributions
L465: Please write out all the names in your acknowledgements and avoid abbreviations (except for OBS). Abbreviations of names appear to devalue the contributions of the very people who stand behind the paper.
We changed that and avoided abbreviations.

---

## Editor Decision (ED1)

**Comments of Editor on the revised manuscript**

The authors have done an admirable job of amending the manuscript to address the major points raised by the two reviewers. The manuscript will be acceptable pending some minor corrections, mostly to improve the semantics and clarity of the text, as well as the clarity of Figure 1. I have marked all suggested changes below in red.

Mark R. Handy

**Responses to Editor**

Abstract:
"The group velocity and shear-wave velocity results compare well to existing large-scale studies that partly include the study area. Onshore France, we observe a high-velocity area beneath the Argentera Massif, roughly 10 km below sea level. We interpret this as the root of the Argentera Massif. Our results add spatial resolution to known seismic velocities in the Ligurian Basin, thereby augmenting existing seismic profiles. In agreement with existing seismic studies, our shear-wave velocity maps indicate a deepening of the Moho from 12 km at the southwestern basin centre to 20-25 km at the Ligurian coast in the northeast and over 30 km at the Provençal coast. The maps also indicate that the southwestern and northeastern Ligurian Basin are structurally separate. The lack of high crustal $v_P/v_S$ ratios beneath the southwestern part of the Ligurian Basin preclude mantle serpentinization there."

L38:
"Today, the Ligurian Sea is 150-225 km wide, whereas the basin itself has a width of 70-170 km (Dannowski et al., 2020), broadening from the northeast to the southwest. The continental margin is narrow (10-20 km) and steep along the Ligurian coast (Finetti et al., 2005) and broader (20-50 km) on the Corsican side (e.g. Rollet et al., 2002).

L317:
"Similarly, the Po Basin has an average sedimentary thickness of 7-8 km "

L352:
"We deduce that for fast areas along the basin axis, the velocity gradient is stronger than away from the basin axis. This is probably caused by the thinning of continental crust (Dannowski et al., 2020) and possible exhumation of denser lower crust and upper mantle rock (Gailler et al., 2009; Jolivet et al., 2015) observed further to the southwest. Both scenarios would lead to a higher S-wave velocity near the basin axis."

L421:
"Rollet et al. (2002) raised the question of whether an offshore prolongation of the Alpine front can be observed onshore France and onshore Corsica. These authors suggested that the Alpine front separates the southwestern and northeastern parts of the Ligurian Basin. This proposed front is roughly located at the boundary between the northeastern and southwestern crustal domains observe in our data (illustrated by the dashed line in Fig. 8e). However, the location and even existence of such a prolongation of the Alpine front beneath the Ligurian sea is not yet resolved. As mentioned above, the seismic records indicate no spreading this far northeast in the basin. Therefore, detection of the proposed offshore Alpine front in the crust is feasible. Dannowski et al. (2020) observe a gradual thickening of the continental crust towards the northeastern Ligurian

Basin. They do not need the sharp step that Makris et al. (1999) introduced between Corsica and the Liguro-Provencal coast to explain the free-air anomaly derived by Sandwell et al. (2014). Our spatial shear-wave velocity data also supports their interpretation. We do not observe a sharp lateral boundary but a gradual change of the velocity layers that fits the model of Dannowski et al. (2020). With the given resolution, an offshore prolongation of the Alpine front is not detectable."

There is an apparent contradiction in the following sentences of the paragraph above: "However, the location and even existence of such a prolongation of the Alpine front beneath the Ligurian sea is not yet resolved. As mentioned above, the seismic records indicate no spreading this far northeast in the basin. Therefore, detection of the proposed offshore Alpine front in the crust is feasible."

First, the authors seem to call into existance the location and even existance of the orogenic front. Then, two sentences later, they say that the detection of this front is possible. Another problem is a logical disconnect between the lack of seismic evidence for spreading in the NE and the existance or non-existance of the Alpine front. The reader is left asking "What does spreading have to do with the front?" They are structures associated with two separate events (spreading in Oligo-Miocene, Alpine orogenic front in Eo-Oligocene) oriented at high angles to each other.

Therefore, I would recommend the following change (which includes the small changes above): "Rollet et al. (2002) raised the question of whether an offshore prolongation of the Alpine front can be observed onshore France and onshore Corsica. These authors suggested that the southwestern and northeastern parts of the Ligurian Basin form, respectively, the footwall and hangingwall of the Alpine front. Thus, the Alpine front would be located approximately at the boundary between the northeastern and southwestern crustal domains distinguished in our data (illustrated by the dashed line in Fig. 8e). Dannowski et al. (2020) observe a gradual thickening of the continental crust towards the northeastern part of theLigurian Basin. They did not need the sharp step that Makris et al. (1999) introduced between Corsica and the Liguro-Provencal coast to explain the free-air anomaly derived by Sandwell et al. (2014). In keeping with Dannowski et al. (2020), our spatial shear-wave velocity data does not show a sharp lateral boundary, but a gradual change of the velocity layers. Detection of an offshore Alpine Front is therefore not feasible with the current resolution."

**Responses to Reviewer 1:**

Introduction:
"The marine bedrock is covered by a sedimentary layer (e.g. Schettino and Turco, 2006) of variable Thickness: less than 3 km thick near the Tuscany coast, increasing towards the southwest to a thickness of up to 8 km offshore Marseille."

L318:
"At up to 9 km depth (Fig. 8a-c), we observe laterally varying shear-wave velocities on land that we assume to be caused by variations in the geology. At the Rhône delta (Fig. 1), where the sedimentary cover is up to 12 km thick (Pichon et al., 2010), we observe $v_s \cong 2.7$ km s$^{-1}$ in the layer at 4-6 km depth (Fig. 8b) and $v_s \cong 3$ km s$^{-1}$ in 6-9 km depth range (Fig. 8c). Similarly, the Po Basin has an average sedimentary thickness of 7-8 km (Molinari et al., 2015a) with shear-wave velocity increasing from $v_s \cong 2.5$ km s$^{-1}$ to $v_s \cong 3.1$ km s$^{-1}$ at 4-9 km depth. In contrast to the sedimentary basins, we observe higher $v_s \cong 3$-3.5 km s$^{-1}$ (4-9 km depth) beneath the Alpine belt, composed of crystalline and metamorphic rocks (e.g. Molinari et al., 2015b). This S-wave variation is most probably caused by the different rocks and structure of the Alpine belt and the sedimentary basins."

Figure 1

Commented [MOU3]: Whose interpretation, Sandwell et al? If so, then write: "…supports the interpretation of Sandwell et al. (2014)."

Commented [MOU4]:

Commented [MOU5]: I removed „but also" because otherwise the sentence makes little sense. One cannot "not observe" in the first part of the sentence, and then "but also observe" in the second part.

Formatted ... [1]

Formatted ... [2]

reaches ... [3]

Commented [MOU6]: You mean the Po Basin filled with sediments, not the Po Plain which is a flat geographical feature.

Formatted ... [4]

Commented [MOU7]: Actually, this is misleading given the tremendous variation in depth of the Po Basin, which ranges from ony 0-1 km along the southern front of the Alps to 9 km beneath the foreland of the Apennines (see isopachs in sheet 1 of the Structural Map of Italy, Bigi et al. 1989). Therefore, the average thickness is closer to 4-5 km. ... [5]

Commented [MOU8]: "Geology" is a generic term for ... [7]

The grey line is barely distinguishable from the background blue of the Ligurian Basin, especially compared to the clear yellow and orange lines. Please use a stronger colour instead of grey, for example, light green.

**Responses to Reviewer 2:**

L76

We decided not to mention the stations that we did not use. Instead, we only focus on the 22 stations that did record all components for the complete deployment, in order to increase the focus of the manuscript. The other stations were not used, because they did not record (enough) data. Please state this in the text explicitly, if not done already.

Section 5.2

"Dannowski et al. (2020) suggest that continental crust was (extremely) thinned along their profile, but that no spreading occurred. This is in-line with our results. A possible spreading centre must be located to the southwest. At the Gulf of Lion margin along the southwestern edge of our research area, Gailler et al. (2009) interpreted their results as oceanic crust, also observing a transition zone made up of "lower crustal material or mixture of serpentinized upper mantle material with lower crustal material" (Gailler et al., 2009). Later, Jolivet et al. (2015) attributed the shallow high velocities to exhumed lower crust and possibly also partially serpentinised mantle. Therefore, a spreading center may have been located southwest of our research area, possibly as close as the Gulf of Lion margin."

| Page 2: [1] Formatted | Microsoft Office User | 13/10/2021 14:07:00 |

English (US)

| Page 2: [1] Formatted | Microsoft Office User | 13/10/2021 14:07:00 |

English (US)

| Page 2: [1] Formatted | Microsoft Office User | 13/10/2021 14:07:00 |

English (US)

| Page 2: [1] Formatted | Microsoft Office User | 13/10/2021 14:07:00 |

English (US)

| Page 2: [1] Formatted | Microsoft Office User | 13/10/2021 14:07:00 |

English (US)

| Page 2: [1] Formatted | Microsoft Office User | 13/10/2021 14:07:00 |

English (US)

| Page 2: [1] Formatted | Microsoft Office User | 13/10/2021 14:07:00 |

English (US)

| Page 2: [1] Formatted | Microsoft Office User | 13/10/2021 14:07:00 |

English (US)

| Page 2: [2] Formatted | Microsoft Office User | 12/10/2021 20:49:00 |

Font colour: Text 1

| Page 2: [2] Formatted | Microsoft Office User | 12/10/2021 20:49:00 |

Font colour: Text 1

| Page 2: [3] Deleted | Microsoft Office User | 12/10/2021 20:50:00 |

| Page 2: [3] Deleted | Microsoft Office User | 12/10/2021 20:50:00 |

| Page 2: [3] Deleted | Microsoft Office User | 12/10/2021 20:50:00 |

| Page 2: [3] Deleted | Microsoft Office User | 12/10/2021 20:50:00 |

| Page 2: [4] Formatted | Microsoft Office User | 13/10/2021 13:12:00 |

Font colour: Text 1

| Page 2: [4] Formatted | Microsoft Office User | 13/10/2021 13:12:00 |

Font colour: Text 1

| Page 2: [5] Commented [MOU7] | Microsoft Office User | 13/10/2021 13:33:00 |

Actually, this is misleading given the tremendous variation in depth of the Po Basin, which ranges from ony 0-1 km along the southern front of the Alps to 9 km beneath the foreland of the Apennines (see isopachs in sheet 1 of the Structural Map of Italy, Bigi et al. 1989). Therefore, the average thickness is closer to 4-5 km. Molinaris larger estimate is valid only for the the S part of the basin.

| Page 2: [6] Deleted | Microsoft Office User | 13/10/2021 13:14:00 |

| Page 2: [6] Deleted | Microsoft Office User | 13/10/2021 13:14:00 |

| Page 2: [6] Deleted | Microsoft Office User | 13/10/2021 13:14:00 |

"Geology" is a generic term for everything from lithology to structure to fluids to the geological history. It's too general. Try to be more specific, in this case, referring to the two characteristics of geology (rock type and structure) that are almost always related to changes in rock physical perpoerties.

"Geology" is a generic term for everything from lithology to structure to fluids to the geological history. It's too general. Try to be more specific, in this case, referring to the two characteristics of geology (rock type and structure) that are almost always related to changes in rock physical perpoerties.

---

## Author Response (AR2)

**Answers to 'comments to the author'**

**Title: 3D crustal structure of the Ligurian Basin revealed by ambient noise tomography using ocean bottom seismometer data**

Dear Topical Editor Mark Handy,

We thank you for the fair and constructive editing and review of our manuscript se-2021-55. We have implemented the suggested linguistic improvements. We address the more detailed comments below. The editor comments are indicated in **black** and our replies are in red colour.

**Commented [MOU2]:** Just a question/suggestion: In this text, clarify what you mean by Sea" and "basin". Are you referring to topography or geology? Geologists often use "Sea" or Ocean" to refer only to the part of a basin underlain by oceanic crust, whereas "basin" is a general term meaning a structural depression that has accumulated sediments.
Thank you for pointing this out. We checked the manuscript and now consequently refer to the Ligurian Basin.

There is an apparent contradiction in the following sentences of the paragraph above:
"However, the location and even existence of such a prolongation of the Alpine front beneath the Ligurian sea is not yet resolved. As mentioned above, the seismic records indicate no spreading this far northeast in the basin. Therefore, detection of the proposed offshore Alpine front in the crust is feasible."
First, the authors seem to call into existence the location and even existence of the orogenic front. Then, two sentences later, they say that the detection of this front is possible. Another problem is a logical disconnect between the lack of seismic evidence for spreading in the NE and the existence or non-existence of the Alpine front. The reader is left asking "What does spreading have to do with the front?" They are structures associated with two separate events (spreading in Oligo-Miocene, Alpine orogenic front in Eo-Oligocene) oriented at high angles to each other. Therefore, I would recommend the following change (which includes the small changes above):
"Rollet et al. (2002) raised the question of whether an offshore prolongation of the Alpine front can be observed onshore France and onshore Corsica. These authors suggested that the southwestern and northeastern parts of the Ligurian Basin form, respectively, the footwall and hanging wall of the Alpine front. Thus, the Alpine front would be located approximately at the boundary between the northeastern and southwestern crustal domains distinguished in our data (illustrated by the dashed line in Fig. 8e). Dannowski et al. (2020) observe a gradual thickening of the continental crust towards the northeastern part of the Ligurian Basin. They did not need the sharp step that Makris et al. (1999) introduced between Corsica and the Liguro-Provencal coast to explain the free-air anomaly derived by Sandwell et al. (2014). In keeping with Dannowski et al. (2020), our spatial shear-wave velocity data does not show a sharp lateral boundary, but a gradual change of the velocity layers. Detection of an offshore Alpine Front is therefore not feasible with the current resolution."

We gratefully follow your recommended changes and rewrote the section accordingly. We applied minor changes to clarify that it was Dannowski et al. (2020) who explained the free-air anomaly derived by Sandwell et al. (2014), not Makris et al. (1999).

**Commented [MOU6]:** You mean the Po Basin filled with sediments, not the Po Plain which is a flat geographical feature.

We agree and applied the change.

**Commented [MOU7]:** Actually, this is misleading given the tremendous variation in depth of the Po Basin, which ranges from only 0- 1 km along the southern front of the Alps to 9 km beneath the foreland of the Apennines (see isopachs in sheet 1 of the Structural Map of Italy, Bigi et al. 1989). Therefore, the average thickness is closer to 4-5 km. Molinari's larger estimate is valid only for the S part of the basin.

As proposed, we now state an average thickness of 4-5 km.

**Commented [MOU8]:** "Geology" is a generic term for everything from lithology to structure to fluids to the geological history. It's too general. Try to be more specific, in this case, referring to the two characteristics of geology (rock type and structure) that are almost always related to changes in rock physical properties.

We applied the suggested changes and now state "*rock types and structure*" instead of "*geology*".

**Figure 1**
The grey line is barely distinguishable from the background blue of the Ligurian Basin, especially compared to the clear yellow and orange lines. Please use a stronger colour instead of grey, for example, light green.

We changed the colour of the grey line to light green.

**Commented:** "*We decided not to mention the stations that we did not use. Instead, we only focus on the 22 stations that did record all components for the complete deployment, in order to increase the focus of the manuscript. The other stations were not used, because they did not record (enough) data.*" Please state this in the text explicitly, if not done already.

Regarding the confusion it caused, when we mentioned the unused stations in an earlier version of the manuscript, we stick to not mentioning them at all. We use all available OBS that recorded for the eight months of the deployment.